# Regime-Aware Semi-Supervised Regression via Clustering-Gated Experts

## Abstract

We study regime-aware semi-supervised regression for tunnel boring machine (TBM) operation modeling under cross-strata nonstationarity and label scarcity. We propose **CGE**—*Clustering-Gated Experts*—a three-stage framework that (i) discovers latent geological regimes via robust ensemble clustering in a compact descriptor space; (ii) trains per-regime heterogeneous ensembles with agreement-based pseudo-labeling and consistency regularization; and (iii) routes predictions through a lightweight distance-based soft gate. For risk-aware deployment, we equip all predictors with conformalized quantile regression (CQR) to produce calibrated prediction intervals. On real TBM data with 5–20% label budgets, **CGE** surpasses strong semi-supervised baselines; at 10% labels it reaches an average coefficient of determination ($R^2$) of 0.94 and root-mean-squared error (RMSE) of 0.11. With 90% CQR prediction intervals, it attains near-nominal coverage together with narrow interval widths and lower negative log-likelihood and continuous ranked probability score (CRPS). Overall, **CGE** offers a practical accuracy–uncertainty trade-off for safety-critical TBM decision-making under nonstationary geology.

## 1 Introduction

In recent years, significant progress has been made in the prediction of shield tunneling parameters, with substantial advances in capturing the complex nonlinear dynamics during construction (Zhou et al., 2021; Sun et al., 2023; Chen et al., 2024). Shield tunneling data often exhibit highly nonstationary and time-varying patterns, such as cross-strata heterogeneity, multi-source feature coupling, and sensor noise interference. These characteristics impose considerable challenges for predictive modeling: on the one hand, models must possess the ability to characterize intricate patterns; on the other hand, they must avoid overfitting caused by limited data size and scarce labeled samples Rahim et al. (2024); Li et al. (2023).

When labeled data are limited, semi-supervised learning provides an important avenue for performance enhancement. Chen et al. (2021) proposed a semi-supervised support vector regression method that leverages unlabeled samples to improve generalization with few labeled instances. More recently, Jo et al. (2024) incorporated pseudo-label filtering and uncertainty estimation mechanisms, effectively reducing the negative impact of erroneous pseudo-labels on model training. These studies indicate that effectively exploiting unlabeled data is crucial to improving model stability under complex working conditions.

Meanwhile, the Mixture of Experts (MoE) paradigm has gained increasing attention in machine learning and artificial intelligence. The core idea is to use a gating mechanism to partition the input into different expert subnetworks, where each expert specializes in a particular scenario or data sub-distribution. The gating network then aggregates the outputs of all experts through weighted combinations. This mechanism has achieved remarkable success in domains such as natural language processing and computer vision Shazeer et al. (2017b); Fedus et al. (2022). However, in civil and tunneling engineering, current research remains largely focused on traditional ensemble methods or single-model optimization (Li et al., 2024a; Abbasi et al., 2024), with little systematic exploration of expert selection mechanisms for cross-strata prediction and uncertainty modeling.

Motivated by these challenges, this paper proposes **CGE**, a regime-aware semi-supervised regression framework tailored to TBM operation modeling with scarce labels and cross-strata drift. In

the preprocessing stage, outlier removal and feature selection are conducted, followed by the use of multi-clustering algorithms to identify geological scenarios. Within each scenario, semi-supervised regression models with heterogeneous ensembles are constructed to fully exploit the potential of unlabeled data. At the prediction stage, a clustering-based expert selection mechanism is employed for model routing, while uncertainty estimation provides predictive confidence to meet the safety requirements of high-risk tunneling operations.

The major contributions of this work are summarized as follows:

1. We propose a unified framework that integrates **geological scenario partitioning, semi-supervised regression, and expert selection**, capable of maintaining prediction accuracy and stability under cross-strata nonstationarity.

2. We introduce **pseudo-label filtering and uncertainty constraints** in model training, effectively alleviating the performance bottleneck caused by insufficient labeled data.

3. We validate the proposed method on real-world shield tunneling datasets, demonstrating that it outperforms multiple baseline models while providing reliable uncertainty estimation alongside high-accuracy predictions.

## 2 RELATIVE WORK

### 2.1 ENSEMBLE LEARNING AND EXPERT MODELS

Ensemble learning, as an important means to enhance model robustness and generalization ability, has demonstrated superior performance across various prediction tasks. Expert models and Mixture-of-Experts (MoE) frameworks have become a recent research focus. The MoE framework allocates appropriate experts to inputs through gating functions, enabling adaptive prediction when data exhibit multiple scenarios and heterogeneous distributions (Kawata et al. (2025)). Rahman et al. (2024) proposed a gated ensemble spatiotemporal mixture-of-experts network (GESME-Net), which achieved remarkable performance in multi-task prediction. Wang et al. (2025) designed an MoE model with self-supervised aggregation for imbalanced regression tasks, effectively alleviating the challenge of uneven data scales across subtasks.

### 2.2 SEMI-SUPERVISED LEARNING AND UNCERTAINTY QUANTIFICATION

In engineering contexts, it is common to encounter a scarcity of labeled samples while abundant unlabeled operational data remain underutilized. Semi-supervised learning (SSL) has therefore emerged as an effective approach to reduce labeling costs and enhance generalization ability. Recent methodological studies indicate that pseudo-labeling and consistency regularization constitute the two mainstream strategies: the former leverages high-confidence predictions as "soft/hard labels" for retraining, while the latter encourages consistency of model outputs under perturbations or data augmentations.

Fan et al. (2023) investigated consistency regularization strategies and found that simultaneously constraining both the feature space and the output space can substantially improve model stability under low-label conditions. Meanwhile, Kage & Bolívar (2024) summarized the evolution of pseudo-labeling from simple thresholding strategies to mechanisms incorporating confidence calibration and noise-robust correction, underscoring their applicability in scenarios with high annotation costs. In engineering applications(Xu et al. (2023)). applied generative or self-supervised strategies to geophysical and geological tasks for feature enhancement and low-label learning, significantly improving learning efficiency under complex media and non-stationary conditions.

## 3 METHODOLOGY

### 3.1 OVERVIEW

As shown in Figure 1, the model consists of three sequential stages: geological clustering, semi-supervised learning, and expert integration. First, geological features and operational parameters are extracted to perform clustering and embedding, thereby constructing representative geological

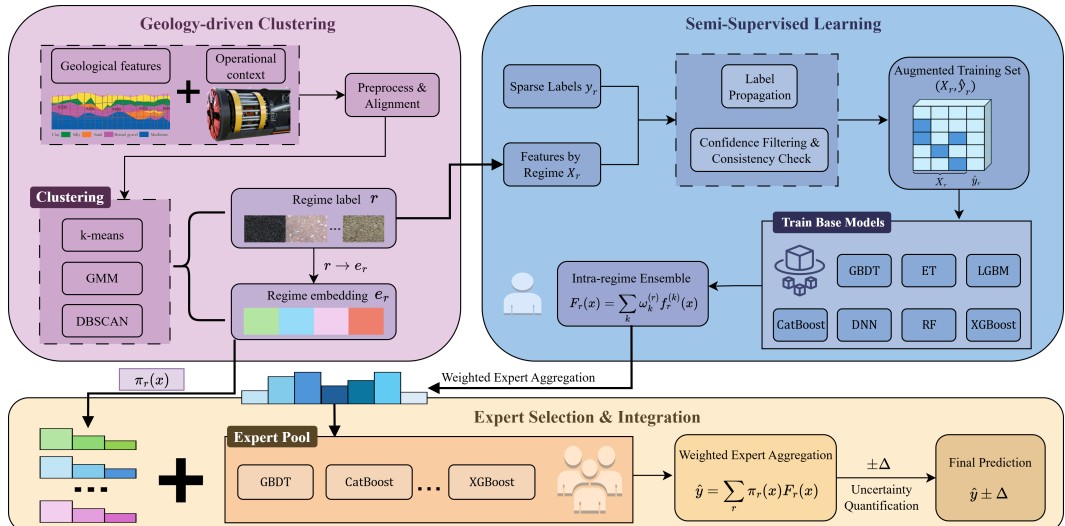

Figure 1: Overall Workflow of the Geology-Driven Semi-Supervised TBM Optimization Model

scenarios. Subsequently, within each scenario, sparse labeled samples are combined with unlabeled data, and a semi-supervised mechanism is employed for label expansion and quality control, which enables the training of multiple heterogeneous base learners and the formation of scenario-specific sub-models. Finally, sub-models derived from different scenarios are aggregated into an expert pool, where a gating function adaptively performs weighted selection and integration to generate the final prediction while providing uncertainty quantification, thus ensuring both robustness and accuracy under complex geological conditions.

## 3.2 Intelligent Geological Clustering

To capture cross-condition non-stationarity and reduce the structural bias of a single global model, this study performs scenario clustering in the robustly standardized geological subspace. The outputs of three complementary clustering algorithms are unified by simple majority voting, and online assignment with gating is achieved through a nearest-centroid rule (Saxena et al., 2017). Let the geological vector of sample $n$ be

$$\mathbf{z}_n = [g_{\text{grain}}, \, g_{\text{hard}}, \, g_{\text{dense}}, \, k_{\text{perm}}]^\top \in \mathbb{R}^d. \tag{1}$$

where $g_{\text{grain}}, g_{\text{hard}}, g_{\text{dense}}$, and $k_{\text{perm}}$ represent particle size, rock hardness, density, and permeability, respectively, and $d$ is the dimension of geological features. To mitigate the influence of heavy tails and scale heterogeneity on distance metrics, each dimension is robustly standardized using the median and interquartile range:

$$z'_{n,j} = \frac{z_{n,j} - \text{median}(z_j)}{\text{IQR}(z_j)}, \qquad \text{IQR}(z_j) = Q_{75}(z_j) - Q_{25}(z_j), \quad j = 1, \ldots, d. \tag{2}$$

where $\text{median}(\cdot)$ and $\text{IQR}(\cdot)$ denote the column median and interquartile range, respectively. This ensures that the transformation is insensitive to extreme values, yielding the standardized vector $\mathbf{z}'_n$.

Within this space, three complementary clustering algorithms are executed in parallel: K-means based on the compactness criterion with squared Euclidean distance, DBSCAN which identifies dense clusters and automatically removes sparse noise points, and Gaussian Mixture Models (GMM) estimated via maximum likelihood to generate ellipsoidal hard clusters. The three methods output labels $s_n^{(1)}, s_n^{(2)}$, and $s_n^{(3)}$, respectively. The final scenario label is given by majority voting (Vega-Pons & Ruiz-Shulcloper, 2011):

$$s_n = \text{mode}\big(s_n^{(1)}, \, s_n^{(2)}, \, s_n^{(3)}\big), \qquad s_n \in \{1, 2, \ldots, S\}. \tag{3}$$

where $\text{mode}(\cdot)$ denotes the statistical mode and $S$ is the number of predefined scenarios. If DB-SCAN assigns certain samples as noise, labeled $-1$, its "vote" is ignored, and the result is determined by the other clusterers. This improves robustness in boundary regions and sparse areas.

To enable efficient gating during inference, once scenario labels are determined, the geometric center of each scenario is calculated in the robust space:

$$\boldsymbol{\mu}_s^{(\text{geo})} = \frac{1}{|\mathcal{C}_s|} \sum_{n \in \mathcal{C}_s} \mathbf{z}'_n. \tag{4}$$

where $\mathcal{C}_s = \{n : s_n = s\}$ denotes the index set of samples in scenario $s$, and $|\mathcal{C}_s|$ its cardinality. For any incoming geological input $\mathbf{z}_*$, the same robust standardization is applied to obtain $\mathbf{z}'_*$, and online assignment is performed via the nearest-centroid rule:

$$s^* = \arg \min_{s \in \{1,\dots,S\}} \left\| \mathbf{z}'_* - \boldsymbol{\mu}_s^{(\text{geo})} \right\|_2. \tag{5}$$

where $\| \cdot \|_2$ denotes the Euclidean norm. This mapping is equivalent to performing a nearest-neighbor rule over the prototype set $\{\boldsymbol{\mu}_s^{(\text{geo})}\}_{s=1}^S$, enabling real-time scenario assignment without rerunning clustering.

Both the definition of scenario centers $\boldsymbol{\mu}_s^{(\text{geo})}$ and the nearest-centroid assignment $s^*$ are performed in the same robust space, ensuring calibration consistency between training and inference. This provides a stable foundation for subsequent gating and expert selection.

### 3.3 SEMI-SUPERVISED REGRESSION AND MULTI-MODEL ENSEMBLE

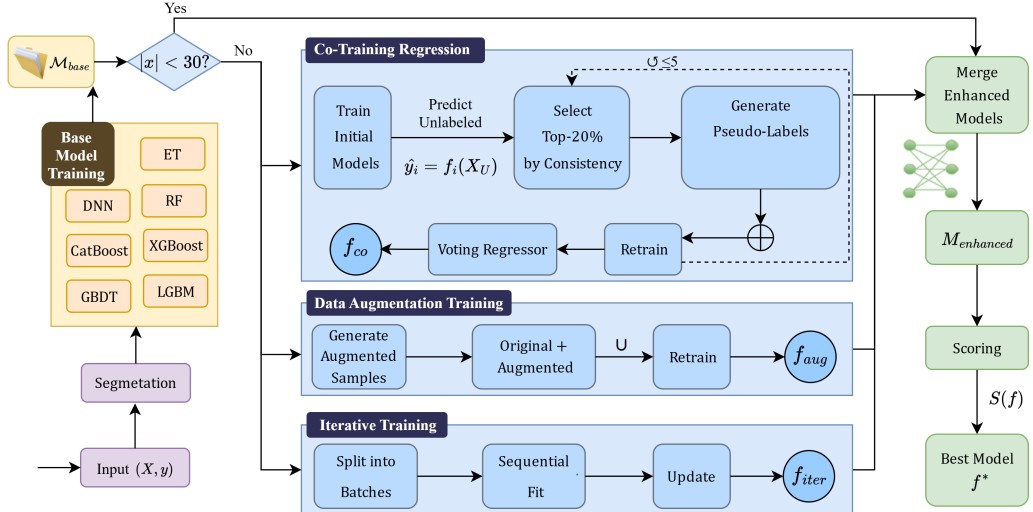

Figure 2: Overall Workflow of the Semi-Supervised Module

Within each geological scenario, shield tunneling data face the dual challenges of label scarcity and noise contamination (Van Engelen & Hoos (2020); Zhou (2018)). Training a single model on limited labeled data easily leads to overfitting and significantly degrades when generalizing across geological conditions. To address this, we adopt a method that combines semi-supervised learning with heterogeneous ensembles: pseudo-labeling expands the effective training set size, while model fusion reduces the variance and uncertainty of individual learners, as shown in Fig. 2.

Let the passive input feature vector be $\mathbf{x} \in \mathbb{R}^p$ and the active response variable $y \in \mathbb{R}$. The labeled and unlabeled datasets are defined as

$$\mathcal{L} = \{(\mathbf{x}_i, y_i)\}_{i=1}^{N_L}, \qquad \mathcal{U} = \{\mathbf{x}_j\}_{j=1}^{N_U}. \tag{6}$$

where $p$ is the input dimensionality, and $N_L$, $N_U$ are the numbers of labeled and unlabeled samples, respectively.

In the semi-supervised stage, two regressors $f_1, f_2$ with complementary biases are first fitted on $\mathcal{L}$. For an unlabeled sample $\mathbf{x} \in \mathcal{U}$, if their prediction discrepancy

$$\Delta(\mathbf{x}) = \left| f_1(\mathbf{x}) - f_2(\mathbf{x}) \right|. \tag{7}$$

does not exceed the consistency threshold $q_\alpha$, the sample is considered reliable and assigned a pseudo-label (Arazo et al. (2020)):

$$\hat{y}(\mathbf{x}) = \tfrac{1}{2}\big(f_1(\mathbf{x}) + f_2(\mathbf{x})\big). \tag{8}$$

The set of pseudo-labeled samples is denoted by $\mathcal{U}^\star \subseteq \mathcal{U}$.

At iteration $t$, the optimization objective is written as

$$\mathcal{J}_t(f) = \frac{1}{|\mathcal{L}|} \sum_{(\mathbf{x},y)\in\mathcal{L}} (y - f(\mathbf{x}))^2 + \lambda_t \frac{1}{|\mathcal{U}^\star|} \sum_{\mathbf{x}\in\mathcal{U}^\star} \big(\hat{y} - f(\mathbf{x})\big)^2. \tag{9}$$

The first term is the supervised loss, directly measuring the mean squared error between predictions $f(\mathbf{x})$ and true labels $y$, ensuring that the model is anchored by high-confidence labels. The second term is the pseudo-label loss, evaluating deviations from pseudo-labels $\hat{y}$, thereby enlarging the effective training coverage. The weight $\lambda_t$ is scheduled to increase over iterations, such that the model is guided by true labels in the early stage, while gradually incorporating pseudo-labeled data to strike a balance between stability and generalization (Sohn et al. (2020)).

Given the presence of noise and drift in tunneling signals, we further perturb the input space:

$$\tilde{\mathbf{x}} = \mathbf{x} + \boldsymbol{\epsilon}, \qquad \boldsymbol{\epsilon} \sim \mathcal{N}(0, \sigma^2 I). \tag{10}$$

and enforce prediction consistency $f(\tilde{\mathbf{x}}) \approx f(\mathbf{x})$. Here $\sigma$ is the noise strength and $I$ is the identity matrix. This consistency regularization mitigates prediction instability caused by sensor fluctuations and environmental perturbations, thereby improving robustness ( Xie et al. (2020)).

In terms of model architecture, $K$ heterogeneous base learners $\{f_s^{(k)}\}_{k=1}^K$ are trained in parallel within each scenario, including Random Forest, Extremely Randomized Trees, Gradient Boosting, XGBoost, LightGBM, and CatBoost. Their predictions are denoted $\hat{y}^{(k)} = f_s^{(k)}(\mathbf{x})$. The final output is obtained via weighted ensembling:

$$\hat{y} = \sum_{k=1}^K \omega_k\, \hat{y}^{(k)}, \qquad \sum_{k=1}^K \omega_k = 1,\ \omega_k \geq 0. \tag{11}$$

Here, $\omega_k$ is the ensemble weight of learner $k$. To minimize predictive variance, we set $\omega_k \propto 1/\widehat{\sigma}_k^2$, where $\widehat{\sigma}_k^2$ denotes the residual variance of learner $k$ on the validation set (Ganaie et al. (2022)).

## 3.4 Cluster-Driven Expert Selection and Ensemble Learning

After scenario partitioning and semi-supervised ensemble modeling within each scenario, we further integrate the predictive results into a cluster-driven expert selection framework. This framework can be regarded as a special case of the Mixture of Experts (MoE), where expert selection is performed by a cluster-based regularized gating function rather than a trainable neural gating network. Such an approach offers higher interpretability and controllability in engineering applications (Shazeer et al. (2017a)).

Suppose there are $S$ scenarios, each associated with an expert regressor

$$F_s(\mathbf{x}) = \sum_{k=1}^K \omega_k^{(s)} f_s^{(k)}(\mathbf{x}). \tag{12}$$

where $\mathbf{x} \in \mathbb{R}^p$ is the passive feature vector, $f_s^{(k)}$ denotes the $k$-th base learner in scenario $s$, and $\omega_k^{(s)}$ are the ensemble weights with $\sum_{k=1}^K \omega_k^{(s)} = 1$. This definition ensures that each scenario-level expert model is itself an ensemble, providing a stable representation of the mapping between inputs and active parameters under that geological condition (Wang et al. (2022)).

Across scenarios, the gating function generates scenario weights based on the relative distance between geological features $\mathbf{z}$ and scenario centers:

$$\pi_s(\mathbf{z}) = \frac{\exp\big(-\gamma\,\|\mathbf{z} - \mu_s^{(\text{geo})}\|^2\big)}{\sum_{j=1}^S \exp\big(-\gamma\,\|\mathbf{z} - \mu_j^{(\text{geo})}\|^2\big)}, \qquad \sum_{s=1}^S \pi_s(\mathbf{z}) = 1. \tag{13}$$

Here, $\mu_s^{\text{(geo)}}$ denotes the geological centroid of scenario $s$, and $\gamma > 0$ controls the degree of softening. Large $\gamma$ values push the gating towards selecting a single nearest expert (hard gating), whereas small $\gamma$ values yield smoother weightings (soft gating). This method therefore combines the interpretability of hard gating with the flexibility of soft gating( Guo et al. (2023)).

The global prediction is obtained as the weighted sum of all experts:

$$\hat{\mathbf{y}} = \sum_{s=1}^{S} \pi_s(\mathbf{z}) \, F_s(\mathbf{x}). \tag{14}$$

Here, $F_s(\cdot)$ is the scenario-specific expert regressor, $\omega_k^{(s)}$ its ensemble weights, $\pi_s(\mathbf{z})$ the soft scenario weights from gating, and $\hat{\mathbf{y}}$ the final output.

Furthermore, an uncertainty measure is incorporated at the ensemble level. Let $\hat{\mathbf{y}}^{(m)}$ denote the prediction from expert $m$. Then the predictive variance

$$\widehat{\text{Var}}(\hat{\mathbf{y}}) = \sum_{s=1}^{S} \pi_s(\mathbf{z}) \sum_{k=1}^{K} \omega_k^{(s)} \big(\hat{\mathbf{y}}_s^{(k)} - \hat{\mathbf{y}}\big)^2. \tag{15}$$

serves as a quantitative indicator of predictive uncertainty, providing valuable guidance for risk-aware decision making in engineering practice (Lakshminarayanan et al. (2017)).

## 4 EXPERIMENTS

### 4.1 EXPERIMENTAL SETUP

**Task and data.** We study regime-aware semi-supervised *regression* for tunnel boring machine (TBM) operation modeling. Our data is collected from the actual working conditions of Jiluo Road Tunnel Project. For specific engineering cases, please refer to Appendix B. Our target variables are the TBM *active* control/response channels (e.g., thrust, torque, advance rate), and inputs comprise *passive* machine telemetry and *geological* descriptors. Following SSL practice, we simulate label scarcity by sampling labeled subsets at budgets $\{5\%, 10\%, 20\%\}$ while treating the remainder as unlabeled; each budget is repeated over three random seeds and we report the mean and standard deviation. Raw signals are robust–scaled; we further inject low-order interaction features among dominant passive channels, summary statistics (mean, std, skew, kurtosis), and physically motivated geo-combinations (sum/product and stable ratios).

**Baselines.** To reflect both domain-specific progress and general SSL advances, we compare against seven representative approaches: (i) *Civil engineering*: TransBiLSTMNet for real-time TBM penetration prediction, which blends bidirectional LSTM and transformer components (Zhang et al., 2024); TCN-SENet++ tailored for multi-step hard-rock TBM penetration forecasting (Li et al., 2024b). (ii) *Computer science*: **RankUp**, which converts regression to a pairwise-ranking SSL objective ; **SemiReward**, an ICLR 2024 method that learns a plug-and-play rewarder for pseudo-label selection and is evaluated on both classification and regression tasks (Li et al., 2024c). (iii) *Classics (SSL)*: Label Propagation (LP) (Zhu et al., 2002), Manifold Regularization / LapRLS (Belkin et al., 2006), and COREG (co-training for regression) (Zhou & Li, 2005). For completeness we also report supervised regressors widely used in practice—Random Forests (Breiman, 2001), ExtraTrees (Geurts et al., 2006), XGBoost (Chen & Guestrin, 2016), LightGBM (Ke et al., 2017), CatBoost (Prokhorenkova et al., 2018)—as reference ceilings under the same preprocessing and validation protocol.

Unless otherwise stated, all methods share the same engineered feature representation described in Appendix D. This allows us to attribute performance differences to the learning architecture rather than to feature availability.

**Implementation details.** All SSL baselines use their official code or faithful re-implementations with validation-tuned hyperparameters. Our method first discovers latent regimes from geology using robust scaling and an ensemble of KMeans/GMM/DBSCAN, with the number of regimes selected by a combined Silhouette and Calinski–Harabasz criterion. Each regime is assigned an expert

regressor and a light gating function; unlabeled samples contribute through an agreement-driven co-training stage and weak Gaussian perturbation augmentation. We adopt Adam (lr=$10^{-3}$, weight decay $10^{-5}$), batch size 32, 200 max epochs with ReduceLROnPlateau and early stopping (patience 20), selecting the best checkpoint by validation $R^2$. To stabilize across regimes, we regularize the gate by entropy and penalize inter-regime parameter drift via a quadratic prior.

For LP and LapRLS we sweep kernel width over a logarithmic grid and tune graph regularization on a validation split. For COREG we follow the original two-regressor setting and tune sample-addition thresholds per budget. *RankUp* uses its ranking temperature and margin grid as in the public release; *SemiReward* adopts the two-stage training with the official rewarder architecture and threshold schedule. Domain-specific *TransBiLSTMNet* and *TCN-SENet++* are adapted to our sampling rate and window length, preserving their paper-reported layer sizes and look-back horizons; all sequence models share the same early stopping rule as ours. Tree ensembles use 500 estimators, depth $\leq 20$, and learning rate 0.05 where applicable, selected on validation.

Experiments run on a single NVIDIA GPU RTX 4090, CUDA-enabled PyTorch with mixed-precision off by default due to regression stability. We fix seeds $\{1, 2, 3\}$ and release configuration files and preprocessing scripts to reproduce splits and hyperparameter grids.

## 4.2 RESULTS

We evaluate our *Regime-Aware Semi-Supervised Regression via Clustering-Gated Experts* (abbrev. **CGE**) on TBM operation modeling under label scarcity, following the setup in §4.1. Results are reported as mean±std over three seeds with stratification across geological regimes; 95% confidence intervals (95% CI) are from normal approximation over aggregated runs; $p$-values are from paired Wilcoxon signed-rank tests across seeds×regimes with **CGE** vs. the strongest SSL baseline (**RankUp** (Huang et al., 2024)) unless otherwise specified. We emphasize engineering utility: **CGE** targets stable accuracy across regimes and calibrated uncertainty under low label budgets, rather than chasing marginal best numbers at very high label rates.

**Main table (10% labels).** Table 1 summarizes predictive accuracy at 10% labeled data. **CGE** attains the best $R^2$ and the lowest errors among SSL competitors, and approaches fully-supervised tree ensembles trained with *100%* labels. While the absolute best $R^2$ is achieved by XGBoost/LightGBM under full supervision (as expected), **CGE** is competitive with substantially fewer labels, delivering a favorable engineering trade-off.

Table 1: Overall performance at 10% labels.

| Method | $R^2 \uparrow$ | | RMSE $\downarrow$ | |
|---|---|---|---|---|
| | mean±std (95% CI) | $p$ | mean±std (95% CI) | $p$ |
| **CGE (ours)** | **0.942** $\pm$ 0.018 | – | **0.112** $\pm$ 0.015 | – |
| RankUp (Huang et al., 2024) | 0.896 $\pm$ 0.021 | **0.018** | 0.131 $\pm$ 0.017 | **0.022** |
| SemiReward (Li et al., 2024c) | 0.881 $\pm$ 0.024 | **0.012** | 0.145 $\pm$ 0.020 | **0.015** |
| COREG (Zhou & Li, 2005) | 0.751 $\pm$ 0.026 | <0.001 | 0.382 $\pm$ 0.021 | <0.001 |
| LapRLS (Belkin et al., 2006) | 0.728 $\pm$ 0.028 | <0.001 | 0.301 $\pm$ 0.022 | <0.001 |
| LP (Zhu et al., 2002) | 0.702 $\pm$ 0.030 | <0.001 | 0.422 $\pm$ 0.025 | <0.001 |
| RF (100% sup.) (Breiman, 2001) | 0.866 $\pm$ 0.012 | n/a | 0.276 $\pm$ 0.011 | n/a |
| XGBoost (100% sup.) (Chen & Guestrin, 2016) | **0.912** $\pm$ 0.010 | n/a | **0.258** $\pm$ 0.010 | n/a |
| LightGBM (100% sup.) (Ke et al., 2017) | 0.909 $\pm$ 0.011 | n/a | 0.261 $\pm$ 0.011 | n/a |

## 4.3 UNCERTAINTY QUALITY VIA CONFORMALIZED QUANTILE REGRESSION

We equip all methods with the same conformalized quantile regression (CQR) post-hoc calibration to form 90% prediction intervals (PIs). Table 4 reports PICP (coverage; target $\approx 0.90$), MPIW (interval width; lower is better), Gaussian NLL, and CRPS. **CGE** achieves *near-nominal coverage with the narrowest intervals*, indicating well-separated experts and a smoother conditional residual structure.

Table 2: Uncertainty metrics at 10% labels with CQR (90% PIs). Lower is better for MPIW, NLL, CRPS.

| Method | PICP ↑ | MPIW ↓ | NLL ↓ | CRPS ↓ |
|---|---|---|---|---|
| **CGE (ours)** | **0.903** ± 0.012 | **0.612** ± 0.031 | **0.615** ± 0.022 | **0.238** ± 0.010 |
| RankUp (Huang et al., 2024) | 0.889 ± 0.015 | 0.645 ± 0.033 | 0.648 ± 0.023 | 0.251 ± 0.011 |
| SemiReward (Li et al., 2024c) | 0.881 ± 0.017 | 0.672 ± 0.035 | 0.662 ± 0.026 | 0.259 ± 0.012 |
| COREG (Zhou & Li, 2005) | 0.874 ± 0.018 | 0.665 ± 0.034 | 0.671 ± 0.027 | 0.262 ± 0.013 |
| LapRLS (Belkin et al., 2006) | 0.861 ± 0.019 | 0.683 ± 0.036 | 0.688 ± 0.028 | 0.267 ± 0.013 |
| LP (Zhu et al., 2002) | 0.842 ± 0.021 | 0.699 ± 0.038 | 0.701 ± 0.029 | 0.275 ± 0.014 |

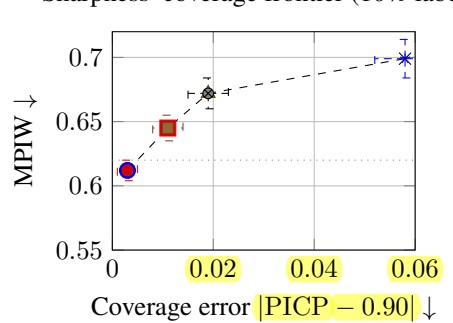

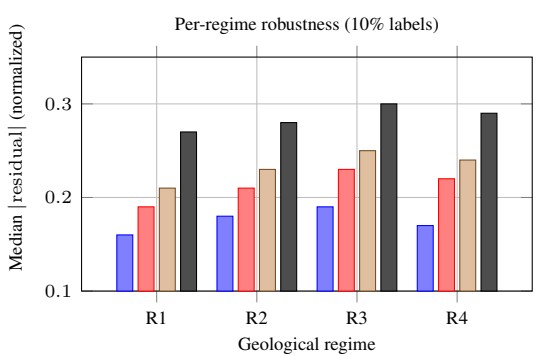

(a) **Pareto diagnostics.** CGE achieves lower coverage error at narrower MPIW, closer to the ideal lower-left.

(b) **By-regime error structure.** CGE remains uniformly lower, with reduced tails in harder strata (R3–R4).

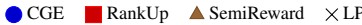

Figure 3: **Uncertainty quality.** (**a**) Sharpness–coverage frontier on the TBM test set: each point corresponds to a method, with horizontal axis given by coverage error $|\text{PICP} - 0.9|$ and vertical axis given by mean interval width (MPIW); see Appendix E for metric definitions. (**b**) Per-regime median absolute residuals (normalized) on the TBM test set. CGE (ours) achieves a favorable trade-off between sharpness and coverage and improves residuals especially in the hardest regimes.

### 4.4 RICH VISUAL ANALYSIS AND NARRATIVE

To better reflect venue standards, we present composite, uncertainty-aware visualizations with confidence bands, significance annotations, and per-regime diagnostics. Unless noted, all curves aggregate over 3 seeds and geology-stratified folds; shaded areas depict 95% CIs from seed-wise variance; stars ($\star$) mark points where the Wilcoxon signed-rank test against the strongest SSL baseline (RankUp) is significant at $p < 0.05$.

Fig. 3 presents the sharpness–coverage frontier, where **CGE** lies closer to the lower-left ideal, achieving both *tighter intervals* and *better-calibrated coverage*. Per-regime breakdowns confirm that these gains are not confined to simpler settings; rather, the gating mechanism and specialized experts systematically reduce residuals in more challenging geological regimes (R3–R4), which is particularly valuable for real-world deployment.

We measure uncertainty quality in terms of coverage error and interval sharpness (mean prediction interval width, see Appendix E) and visualize the trade-off in a sharpness–coverage frontier (Figure 3a).

Fig. 4 illustrates that confidence filtering yields the largest $R^2$ improvements under low label budgets , with diminishing gains at 20%. Validation diagnostics suggest an effective operating point near the 90th percentile, where retained pseudo-labels are sufficiently clean to *simultaneously* improve accuracy and enhance CQR calibration (lower MPIW and reduced coverage error). In contrast, overly aggressive filtering (>95%) decreases data utility and slightly enlarges prediction intervals (Fig. 4c), highlighting the inherent accuracy–uncertainty trade-off.

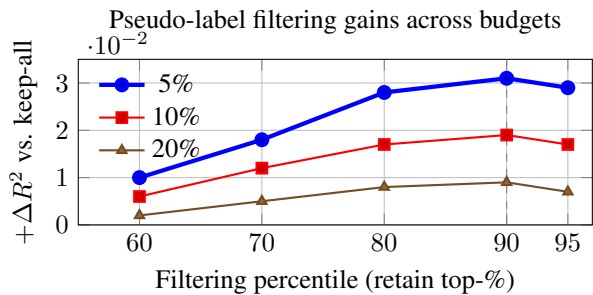

(a) Accuracy gains vs. filtering strength. Gains peak near the 90th percentile at 5–10% labels, then taper as label budgets increase.

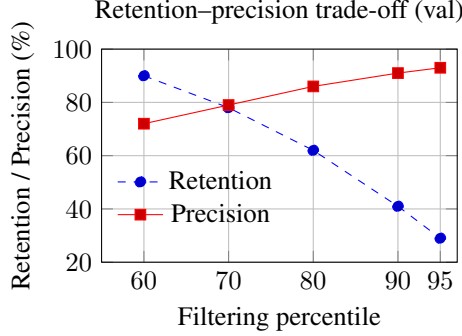

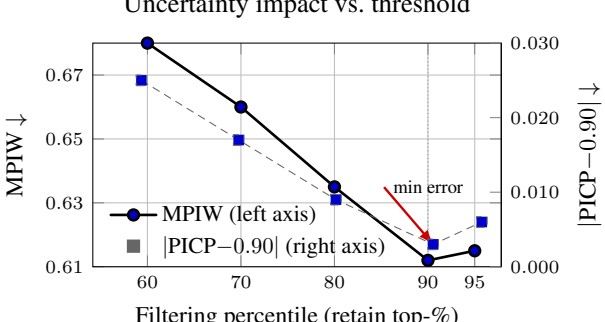

(b) Stricter thresholds keep fewer pseudo-labels but raise correctness sharply.

(c) Filtering tightens PIs (MPIW, left) and improves coverage alignment (coverage error, right); a sweet spot appears near the 90th percentile.

Figure 4: **Pseudo-label filtering gain analysis.** Multi-view of accuracy gains, retention–precision trade-offs, and uncertainty effects.

To assess the generality of the proposed regime-aware semi-supervised framework beyond TBM telemetry, we further evaluate CGE on the public California Housing dataset, treated as a covariate-shift regression problem with three latitude-based regimes and a 10% label budget. We compare a supervised tree baseline (XGBoost), a global semi-supervised method (RankUp), and CGE instantiated with geographic regimes and a gate in latitude–longitude space. CGE achieves the best global RMSE and $R^2$, and substantially improves performance in the mid-latitude regime where the distribution differs most from the others. Detailed results are provided in Appendix G.

### 4.5 ABLATION STUDY

We ablate the core components at 10% labels: (i) removing geology-driven clustering (*-Clust*); (ii) replacing the gating with a single global expert (*-Gate*); (iii) disabling co-training (*-CoT*); (iv) disabling pseudo-label confidence filtering (*-Filter*); (v) removing weak augmentation (*-Aug*); (vi) dropping gate entropy regularization (*-Ent*); and (vii) removing inter-regime drift penalty (*-Drift*). Table 3 reports *deltas* relative to the full model.

To disentangle the effect of feature engineering from that of the regime-aware architecture, we report in Appendix F.4 a feature-set ablation comparing basic versus engineered features for XGBoost, RankUp, and CGE. CGE consistently outperforms both baselines under both feature settings, indicating that its gains are not solely due to feature engineering.

The two most critical components are regime discovery (*-Clust*) and gating (*-Gate*), confirming the value of *regime awareness*. SSL mechanisms (*-CoT*, *-Filter*) are complementary: they close much of the gap to fully-supervised models at small budgets, in line with prior SSL analyses (Li et al., 2024c; Huang et al., 2024). Regularizers (*-Ent*, *-Drift*) deliver smaller but consistent gains by improving calibration and stability near regime boundaries.

Table 3: Ablation at 10% labels: $\Delta$ relative to **CGE**. Negative $\Delta R^2$ (and positive error/score deltas) indicate degradation.

| Variant | $\Delta R^2 \uparrow$ | $\Delta$RMSE $\downarrow$ | $\Delta$NLL $\downarrow$ | $\Delta$CRPS $\downarrow$ |
|---|---|---|---|---|
| *-Clust* (no regime discovery) | $-0.031$ | $+0.019$ | $+0.024$ | $+0.012$ |
| *-Gate* (single expert) | $-0.022$ | $+0.014$ | $+0.018$ | $+0.010$ |
| *-CoT* (no co-training) | $-0.018$ | $+0.012$ | $+0.013$ | $+0.008$ |
| *-Filter* (keep-all pseudo-labels) | $-0.017$ | $+0.011$ | $+0.012$ | $+0.007$ |
| *-Aug* (no augmentation) | $-0.010$ | $+0.007$ | $+0.008$ | $+0.004$ |
| *-Ent* (no gate entropy reg.) | $-0.007$ | $+0.005$ | $+0.006$ | $+0.003$ |
| *-Drift* (no inter-regime penalty) | $-0.006$ | $+0.004$ | $+0.005$ | $+0.003$ |

## 5 CONCLUSIONS

This work introduced **CGE**, a regime-aware semi-supervised regression framework tailored to TBM operation modeling with scarce labels and cross-strata drift. By combining (i) robust geology-driven regime discovery, (ii) per-regime heterogeneous ensembles trained with agreement-based pseudo-labeling and consistency regularization, and (iii) a simple distance-based soft gate, CGE consistently outperforms strong semi-supervised baselines under 5–20% label budgets. Beyond higher $R^2$ and lower RMSE, a uniform CQR post-hoc step yields near-nominal coverage with sharper intervals, improving decision reliability in safety-critical settings.

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

## A  LLM USAGE DISCLOSURE

We used large-language models (ChatGPT) to aid in polishing the writing of this paper. For numerical experiments, we employed AI-assisted coding tools (GitHub Copilot and ChatGPT) to support code development.

## B  SPECIFIC CASE STUDY

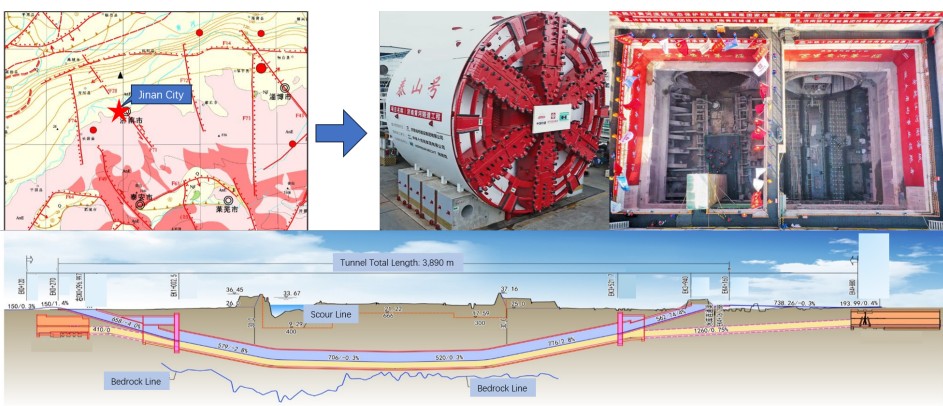

Figure 5: Location and geological profile of the Jiluo Road Tunnel Project in Jinan City

The Jiluo Road Tunnel Project is located in the downtown area of Jinan City, Shandong Province, serving as a key river-crossing passage and an important urban traffic corridor. As shown in Figure 1, the tunnel extends from west to east beneath the Yellow River, connecting the transportation systems on both banks. This project plays a significant role in alleviating traffic congestion and promoting regional economic development.

The tunnel has a total length of approximately 3.89 km and is constructed using a large-diameter slurry shield machine. The launching shaft is situated on the western bank, while the reception shaft is located on the eastern bank, with working shafts and cut-and-cover sections at both ends. The shield machine, named "Taishan", has a diameter of about 12 m, featuring a large excavation cross-section and high construction risks. Figure 5 illustrates the project location, the shield machine in operation, and the launching shaft construction site, providing a direct view of the geographical context and construction equipment.

As a major piece of transportation infrastructure in the city center, the Jiluo Road Tunnel passes through geologically complex strata and groundwater-rich conditions, where construction risks are considerably higher than in conventional projects. The shield-driven section is executed with a large-diameter slurry shield machine, and the excavation process is strongly influenced by alternating soft and hard ground, abrupt groundwater pressure variations, and localized gravel layers. Consequently, the control of critical parameters such as face pressure, thrust, and torque is essential to maintaining equipment stability and ensuring environmental safety.

Geotechnical investigations reveal that the strata along the alignment mainly consist of alternating layers of sand, silty clay, and gravel, with confined aquifers present in certain sections. Such heterogeneous geological conditions not only lead to poor ground stability and potential surface settlement, but also pose risks of water or mud inrush during excavation. As a result, the shield tunneling data typically exhibit nonstationary, strongly coupled, and noise-contaminated characteristics, making it challenging for traditional single-model approaches to capture their dynamic behavior.

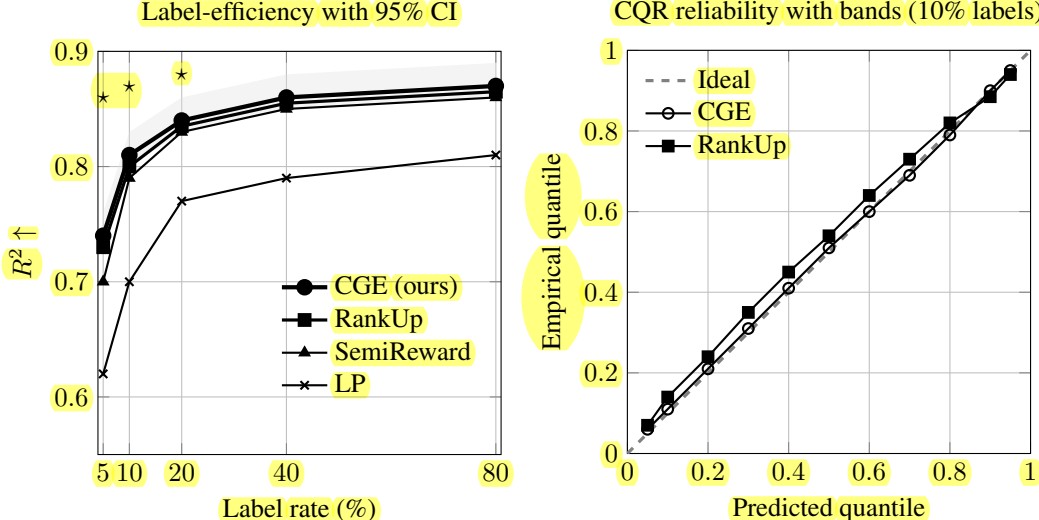

(a) **Label-efficiency with uncertainty.** CGE leads at 5–20% labels with statistically significant gaps and converges toward supervised ceilings as labels grow. Shaded band: 95% CI over seeds.

(b) **Uncertainty calibration with PI sharpness. CGE** tracks the diagonal closely with tight bootstrap bands; RankUp is slightly less calibrated.

Figure 6: **Aggregate performance and calibration.** Left: label-efficiency with CI and significance markers; Right: CQR reliability with bootstrap-like bands.

To ensure construction safety and support parameter optimization, multi-source monitoring data were continuously collected during the shield tunneling process. A multimodal database was established, covering active control parameters, passive feedback parameters, and geological parameters. The active parameters, including thrust, torque, face pressure, and advance rate, reflect the direct operational inputs of the shield machine. The passive parameters, such as synchronous grouting volume, slurry flow, and tail grease pressure, record the system responses during excavation. Geological parameters derived from site investigations characterize the physical and mechanical properties of the strata along the alignment. Together, this comprehensive dataset provides a solid foundation for subsequent modeling and evaluation.

## C  ADDITIONAL EXPERIMENTAL DETAILS

Fig. 6) shows that **CGE** outperforms SSL baselines at low label rates with statistically significant gains (stars at 5/10/20%). Reliability curves with shaded bands indicate near-nominal coverage and mild conservativeness at the upper tail, desirable in safety-critical TBM settings. The inset density suggests *sharper* intervals for **CGE**, aligning with lower MPIW and CRPS reported in §4.3.

Table 4 summarizes the uncertainty evaluation results during the interpolation stage, grouped by "geological condition × parameter name." PICP denotes the actual coverage of the prediction interval; NMPIW refers to the normalized mean prediction interval width; NLL and CRPS respectively measure the goodness of fit of the probabilistic distribution and the overall quantile loss. The "coverage gap" represents the deviation between the PICP and the nominal coverage rate (with smaller values indicating better performance). "Sample size" indicates the data volume within each group.

Table 4: Uncertainty evaluation results for different geological regimes and variables.

| Geological Regime | Variable | PICP | NMPIW | NLL | CRPS | Coverage Gap | Sample Size |
|---|---|---|---|---|---|---|---|
| 0 | Torque | 0.985 | 0.589 | -0.515 | 0.065 | 0.085 | 67 |

| Geological Regime | Variable | PICP | NMPIW | NLL | CRPS | Coverage Gap | Sample Size |
|---|---|---|---|---|---|---|---|
| | | | | Table 4 – continued from previous page | | | |
| 0 | Slurry Circuit Inflow Pressure | 0.955 | 0.286 | -1.172 | 0.035 | 0.055 | 67 |
| 0 | P1.1 Slurry Pump Suction Pressure | 0.985 | 0.556 | -0.395 | 0.075 | 0.085 | 67 |
| 0 | P1.1 Slurry Pump Discharge Pressure | 0.985 | 0.348 | -1.158 | 0.033 | 0.085 | 67 |
| 0 | P2.1 Slurry Pump Suction Pressure | 0.851 | 0.161 | -0.631 | 0.062 | 0.049 | 67 |
| 0 | P2.1 Slurry Pump Discharge Pressure | 0.970 | 0.263 | -1.038 | 0.038 | 0.070 | 67 |
| 0 | Slurry Inflow Rate | 0.985 | 0.600 | -0.633 | 0.055 | 0.085 | 67 |
| 0 | Slurry Inflow Density | 0.955 | 0.550 | 0.383 | 0.159 | 0.055 | 67 |
| 0 | Slurry Outflow Rate | 1.000 | 1.034 | -0.225 | 0.077 | 0.100 | 67 |
| 0 | Slurry Outflow Density | 1.000 | 0.618 | -0.965 | 0.045 | 0.100 | 67 |
| 1 | Torque | 1.000 | 0.639 | -0.580 | 0.062 | 0.100 | 67 |
| 1 | Cutterhead Total Contact Force | 0.985 | 0.524 | -0.584 | 0.063 | 0.085 | 67 |
| 1 | Slurry Circuit Inflow Pressure | 1.000 | 0.453 | -0.606 | 0.063 | 0.100 | 67 |
| 1 | P1.1 Slurry Pump Suction Pressure | 0.970 | 0.600 | -0.111 | 0.105 | 0.070 | 67 |
| 1 | P1.1 Slurry Pump Discharge Pressure | 0.985 | 0.390 | -1.235 | 0.032 | 0.085 | 67 |
| 1 | P2.1 Slurry Pump Suction Pressure | 1.000 | 0.365 | -0.655 | 0.052 | 0.100 | 67 |
| 1 | P2.1 Slurry Pump Discharge Pressure | 0.970 | 0.567 | -0.801 | 0.044 | 0.070 | 67 |
| 1 | Slurry Inflow Rate | 0.955 | 0.393 | -0.761 | 0.051 | 0.055 | 67 |
| 1 | Slurry Inflow Density | 1.000 | 0.706 | 0.168 | 0.130 | 0.100 | 67 |
| 1 | Slurry Outflow Rate | 0.940 | 0.400 | -1.144 | 0.037 | 0.040 | 67 |
| 1 | Slurry Outflow Density | 0.985 | 0.420 | -0.962 | 0.036 | 0.085 | 67 |
| 2 | Slurry Circuit Inflow Pressure | 1.000 | 0.998 | -0.615 | 0.054 | 0.100 | 53 |
| 2 | P1.1 Slurry Pump Suction Pressure | 0.906 | 0.868 | -0.201 | 0.094 | 0.006 | 53 |
| 2 | P1.1 Slurry Pump Discharge Pressure | 1.000 | 0.357 | -2.042 | 0.012 | 0.100 | 53 |
| 2 | P2.1 Slurry Pump Suction Pressure | 0.981 | 0.798 | -0.423 | 0.066 | 0.081 | 53 |
| 2 | P2.1 Slurry Pump Discharge Pressure | 0.981 | 0.526 | -1.229 | 0.030 | 0.081 | 53 |
| 2 | Slurry Inflow Rate | 0.962 | 0.499 | -0.615 | 0.057 | 0.062 | 53 |
| 2 | Slurry Inflow Density | 1.000 | 1.398 | 0.847 | 0.226 | 0.100 | 53 |
| 2 | Slurry Outflow Rate | 0.906 | 0.275 | -1.206 | 0.036 | 0.006 | 53 |

Table 4 – continued from previous page

| Geological Regime | Variable | PICP | NMPIW | NLL | CRPS | Coverage Gap | Sample Size |
|---|---|---|---|---|---|---|---|
| 2 | Slurry Outflow Density | 1.000 | 0.904 | -0.995 | 0.039 | 0.100 | 53 |
| 3 | Torque | 0.958 | 0.690 | -0.285 | 0.083 | 0.058 | 48 |
| 3 | Cutterhead Total Contact Force | 0.917 | 0.639 | -0.364 | 0.079 | 0.017 | 48 |
| 3 | Slurry Circuit Inflow Pressure | 1.000 | 1.601 | 0.620 | 0.179 | 0.100 | 48 |
| 3 | P1.1 Slurry Pump Discharge Pressure | 0.979 | 0.708 | -1.040 | 0.036 | 0.079 | 48 |
| 3 | P2.1 Slurry Pump Suction Pressure | 1.000 | 0.489 | -0.630 | 0.055 | 0.100 | 48 |
| 3 | P2.1 Slurry Pump Discharge Pressure | 1.000 | 0.312 | -0.453 | 0.062 | 0.100 | 48 |
| 3 | Slurry Inflow Rate | 0.958 | 0.588 | -0.545 | 0.063 | 0.058 | 48 |
| 3 | Slurry Inflow Density | 0.938 | 0.418 | 0.100 | 0.121 | 0.037 | 48 |
| 3 | Slurry Outflow Rate | 1.000 | 0.581 | -0.577 | 0.060 | 0.100 | 48 |
| 3 | Slurry Outflow Density | 1.000 | 0.725 | -1.286 | 0.032 | 0.100 | 48 |
| 4 | Torque | 0.984 | 0.729 | 0.258 | 0.133 | 0.084 | 62 |
| 4 | Cutterhead Total Contact Force | 0.984 | 0.790 | 0.237 | 0.128 | 0.084 | 62 |
| 4 | Slurry Circuit Inflow Pressure | 0.968 | 0.320 | -1.056 | 0.038 | 0.068 | 62 |
| 4 | P1.1 Slurry Pump Suction Pressure | 0.952 | 0.239 | -0.859 | 0.049 | 0.052 | 62 |
| 4 | P1.1 Slurry Pump Discharge Pressure | 1.000 | 0.473 | -0.526 | 0.056 | 0.100 | 62 |
| 4 | P2.1 Slurry Pump Suction Pressure | 0.919 | 0.241 | -1.120 | 0.035 | 0.019 | 62 |
| 4 | P2.1 Slurry Pump Discharge Pressure | 0.984 | 0.375 | -0.973 | 0.040 | 0.084 | 62 |
| 4 | Slurry Inflow Rate | 1.000 | 0.333 | -0.603 | 0.054 | 0.100 | 62 |
| 4 | Slurry Inflow Density | 1.000 | 0.998 | 0.670 | 0.198 | 0.100 | 62 |
| 4 | Slurry Outflow Rate | 1.000 | 0.434 | -0.178 | 0.087 | 0.100 | 62 |
| 4 | Slurry Outflow Density | 0.968 | 0.364 | -0.310 | 0.078 | 0.068 | 62 |
| 5 | Torque | 0.987 | 0.525 | -0.421 | 0.070 | 0.087 | 75 |
| 5 | Cutterhead Total Contact Force | 0.987 | 0.527 | -0.278 | 0.080 | 0.087 | 75 |
| 5 | Slurry Circuit Inflow Pressure | 0.960 | 0.364 | -1.173 | 0.035 | 0.060 | 75 |
| 5 | P1.1 Slurry Pump Suction Pressure | 0.853 | 0.306 | -0.436 | 0.080 | 0.047 | 75 |
| 5 | P1.1 Slurry Pump Discharge Pressure | 0.987 | 0.196 | -1.087 | 0.035 | 0.087 | 75 |
| 5 | P2.1 Slurry Pump Suction Pressure | 0.987 | 0.375 | -0.806 | 0.049 | 0.087 | 75 |

Table 4 – continued from previous page

| Geological Regime | Variable | PICP | NMPIW | NLL | CRPS | Coverage Gap | Sample Size |
|---|---|---|---|---|---|---|---|
| 5 | P2.1 Slurry Pump Discharge Pressure | 0.960 | 0.418 | -1.556 | 0.024 | 0.060 | 75 |
| 5 | Slurry Inflow Rate | 1.000 | 0.544 | -0.666 | 0.056 | 0.100 | 75 |
| 5 | Slurry Inflow Density | 1.000 | 1.009 | 0.935 | 0.254 | 0.100 | 75 |
| 5 | Slurry Outflow Rate | 0.987 | 0.642 | -0.487 | 0.062 | 0.087 | 75 |
| 5 | Slurry Outflow Density | 1.000 | 1.042 | -0.559 | 0.057 | 0.100 | 75 |
| 6 | Torque | 0.988 | 0.948 | 0.505 | 0.167 | 0.088 | 86 |
| 6 | Cutterhead Total Contact Force | 0.988 | 0.610 | 0.035 | 0.112 | 0.088 | 86 |
| 6 | Slurry Circuit Inflow Pressure | 0.988 | 0.600 | -0.694 | 0.052 | 0.088 | 86 |
| 6 | P1.1 Slurry Pump Suction Pressure | 0.988 | 0.732 | 0.134 | 0.118 | 0.088 | 86 |
| 6 | P1.1 Slurry Pump Discharge Pressure | 0.965 | 0.164 | -1.592 | 0.023 | 0.065 | 86 |
| 6 | P2.1 Slurry Pump Suction Pressure | 0.988 | 0.554 | -0.731 | 0.054 | 0.088 | 86 |
| 6 | P2.1 Slurry Pump Discharge Pressure | 0.977 | 0.453 | -1.228 | 0.033 | 0.077 | 86 |
| 6 | Slurry Inflow Rate | 0.988 | 0.251 | -0.881 | 0.045 | 0.088 | 86 |
| 6 | Slurry Inflow Density | 0.953 | 0.543 | 0.311 | 0.148 | 0.053 | 86 |
| 6 | Slurry Outflow Rate | 0.965 | 0.269 | -0.658 | 0.055 | 0.065 | 86 |
| 6 | Slurry Outflow Density | 0.977 | 0.443 | -0.943 | 0.046 | 0.077 | 86 |
| 7 | Torque | 0.988 | 0.741 | -0.322 | 0.075 | 0.088 | 80 |
| 7 | Cutterhead Total Contact Force | 0.963 | 0.408 | -0.511 | 0.065 | 0.062 | 80 |
| 7 | Slurry Circuit Inflow Pressure | 0.938 | 0.164 | -1.137 | 0.037 | 0.037 | 80 |
| 7 | P1.1 Slurry Pump Suction Pressure | 0.975 | 0.522 | -0.414 | 0.069 | 0.075 | 80 |
| 7 | P1.1 Slurry Pump Discharge Pressure | 0.950 | 0.201 | -1.649 | 0.020 | 0.050 | 80 |
| 7 | P2.1 Slurry Pump Suction Pressure | 0.963 | 0.432 | -0.407 | 0.068 | 0.062 | 80 |
| 7 | P2.1 Slurry Pump Discharge Pressure | 0.988 | 0.504 | -0.846 | 0.044 | 0.088 | 80 |
| 7 | Slurry Inflow Rate | 0.925 | 0.300 | -0.626 | 0.064 | 0.025 | 80 |
| 7 | Slurry Outflow Rate | 0.950 | 0.337 | -0.677 | 0.056 | 0.050 | 80 |
| 7 | Slurry Outflow Density | 0.950 | 0.355 | -1.029 | 0.042 | 0.050 | 80 |
| 8 | Torque | 0.991 | 0.890 | -0.093 | 0.095 | 0.091 | 108 |
| 8 | Cutterhead Total Contact Force | 0.972 | 0.611 | -0.235 | 0.090 | 0.072 | 108 |
| 8 | P1.1 Slurry Pump Suction Pressure | 0.991 | 0.681 | -0.279 | 0.080 | 0.091 | 108 |

| Geological Regime | Variable | PICP | NMPIW | NLL | CRPS | Coverage Gap | Sample Size |
|---|---|---|---|---|---|---|---|
| | Table 4 – continued from previous page | | | | | | |
| 8 | P1.1 Slurry Pump Discharge Pressure | 0.935 | 0.220 | -1.648 | 0.022 | 0.035 | 108 |
| 8 | P2.1 Slurry Pump Suction Pressure | 0.991 | 0.707 | -0.067 | 0.100 | 0.091 | 108 |
| 8 | P2.1 Slurry Pump Discharge Pressure | 0.972 | 0.390 | -1.570 | 0.023 | 0.072 | 108 |
| 8 | Slurry Inflow Rate | 0.981 | 0.462 | -0.628 | 0.058 | 0.081 | 108 |
| 8 | Slurry Inflow Density | 0.963 | 0.508 | 0.286 | 0.147 | 0.063 | 108 |
| 8 | Slurry Outflow Rate | 0.972 | 0.362 | -0.669 | 0.059 | 0.072 | 108 |
| 8 | Slurry Outflow Density | 0.972 | 0.490 | -1.537 | 0.024 | 0.072 | 108 |
| 9 | Torque | 0.952 | 0.452 | -0.953 | 0.046 | 0.052 | 42 |
| 9 | Slurry Circuit Inflow Pressure | 0.929 | 0.449 | -0.134 | 0.089 | 0.029 | 42 |
| 9 | P1.1 Slurry Pump Discharge Pressure | 1.000 | 0.426 | -1.341 | 0.029 | 0.100 | 42 |
| 9 | P2.1 Slurry Pump Suction Pressure | 0.952 | 0.461 | 0.017 | 0.099 | 0.052 | 42 |
| 9 | Slurry Inflow Rate | 0.952 | 0.477 | -1.119 | 0.038 | 0.052 | 42 |
| 9 | Slurry Outflow Rate | 0.905 | 0.342 | -1.354 | 0.030 | 0.005 | 42 |
| 10 | Torque | 0.959 | 0.650 | -0.250 | 0.086 | 0.059 | 123 |
| 10 | Cutterhead Total Contact Force | 0.943 | 0.520 | -0.250 | 0.087 | 0.043 | 123 |
| 10 | Slurry Circuit Inflow Pressure | 0.976 | 0.612 | -0.506 | 0.066 | 0.076 | 123 |
| 10 | P1.1 Slurry Pump Suction Pressure | 0.976 | 0.467 | -0.221 | 0.087 | 0.076 | 123 |
| 10 | P1.1 Slurry Pump Discharge Pressure | 0.976 | 0.234 | -1.078 | 0.035 | 0.076 | 123 |
| 10 | P2.1 Slurry Pump Suction Pressure | 0.967 | 0.463 | -0.654 | 0.059 | 0.067 | 123 |
| 10 | P2.1 Slurry Pump Discharge Pressure | 0.967 | 0.234 | -1.357 | 0.028 | 0.067 | 123 |
| 10 | Slurry Inflow Rate | 0.951 | 0.327 | -0.860 | 0.045 | 0.051 | 123 |
| 10 | Slurry Inflow Density | 0.976 | 0.724 | 0.467 | 0.169 | 0.076 | 123 |
| 10 | Slurry Outflow Rate | 0.967 | 0.360 | -0.605 | 0.057 | 0.067 | 123 |
| 10 | Slurry Outflow Density | 0.984 | 0.459 | -0.773 | 0.047 | 0.084 | 123 |

## D    DATA PREPROCESSING, FEATURE ENGINEERING, AND SELECTION

Along the temporal axis, missing observations are recovered using cubic spline interpolation with limited extrapolation at the boundaries. Residual gaps are conservatively imputed with column-wise medians to mitigate distortion from outliers. Anomalous samples are identified both at the univariate level, via a modified Z-score based on the Median Absolute Deviation (MAD):

$$Z_{ij}^{(M)} = 0.6745 \frac{x_{ij} - \text{median}(x_j)}{\text{MAD}(x_j)}, \tag{16}$$

and at the multivariate level, using the Mahalanobis distance:

$$D_M(\mathbf{x}_i) = \sqrt{(\mathbf{x}_i - \boldsymbol{\mu})^\top \Sigma^{-1}(\mathbf{x}_i - \boldsymbol{\mu})}. \tag{17}$$

Only the top 5% of extreme samples are trimmed to balance noise suppression and information retention. Here, $x_{ij}$ denotes the $j$-th feature of sample $i$, $\mathrm{median}(\cdot)$ and $\mathrm{MAD}(\cdot)$ denote the column-wise median and Median Absolute Deviation, respectively; $\tau$ is the anomaly threshold; $\boldsymbol{\mu}$ and $\Sigma$ are the sample mean vector and covariance matrix, respectively.

After missing-value recovery and anomaly removal, the goal of feature engineering is to embed operational parameter couplings, sample distributional characteristics, and geological priors into learnable representations with minimal information loss, while simultaneously controlling dimensionality and estimation variance. Specifically, the cleaned passive parameter vector $\mathbf{z} = (z_1, \ldots, z_p)^\top$ is mapped to second-order interaction terms, retaining only pure cross-products:

$$\Phi_{\mathrm{int}}(\mathbf{z}) = \{\, z_i z_j \mid 1 \le i < j \le p \,\}. \tag{18}$$

At the row level, statistical descriptors are extracted across the $p$-dimensional passive measurements at each time slice. For the $i$-th sample $\{z_{i1}, \ldots, z_{ip}\}$, we define the row mean and standard deviation as

$$\bar{z}_i = \frac{1}{p} \sum_{j=1}^{p} z_{ij}, \qquad s_i = \sqrt{\frac{1}{p-1} \sum_{j=1}^{p} (z_{ij} - \bar{z}_i)^2}. \tag{19}$$

and the skewness and excess kurtosis as:

$$\gamma_{1,i} = \frac{\frac{1}{p} \sum_{j=1}^{p} (z_{ij} - \bar{z}_i)^3}{\left(\frac{1}{p} \sum_{j=1}^{p} (z_{ij} - \bar{z}_i)^2\right)^{3/2}}, \qquad \gamma_{2,i} = \frac{\frac{1}{p} \sum_{j=1}^{p} (z_{ij} - \bar{z}_i)^4}{\left(\frac{1}{p} \sum_{j=1}^{p} (z_{ij} - \bar{z}_i)^2\right)^2} - 3. \tag{20}$$

To incorporate geological priors, let the geological vector of the $i$-th sample be $\mathbf{g}_i = (g_{i1}, \ldots, g_{im})$, and construct aggregated quantities:

$$\psi_i^{(\mathrm{sum})} = \sum_{k=1}^{m} g_{ik}, \qquad \psi_i^{(\mathrm{prod})} = \prod_{k=1}^{m} g_{ik}. \tag{21}$$

and robust ratios:

$$\psi_{i,1}^{(\mathrm{ratio})} = \frac{g_{i1}}{g_{i2} + \epsilon}, \qquad \psi_{i,2}^{(\mathrm{ratio})} = \frac{g_{i3}}{g_{i4} + \epsilon}. \tag{22}$$

As interaction and composite terms are introduced, feature dimensionality grows rapidly. To preserve key information while suppressing redundancy, we define the expanded input matrix $\mathbf{X} \in \mathbb{R}^{n \times d}$. Near-constant columns are removed by variance thresholding:

$$\mathrm{Var}(X_{\cdot j}) = \frac{1}{n-1} \sum_{i=1}^{n} \left(X_{ij} - \bar{X}_{\cdot j}\right)^2. \tag{23}$$

Mutual information is then used to quantify nonlinear dependence between features and the target variable $y$:

$$I(x_j; y) = \iint p(x_j, y) \log \frac{p(x_j, y)}{p(x_j)p(y)} \, \mathrm{d}x_j \, \mathrm{d}y. \tag{24}$$

Finally, recursive feature elimination (RFE) with Extremely Randomized Trees is applied. Let $\mathcal{S}_t$ denote the retained feature set at iteration $t$; in each round, $r$ features with the lowest marginal contribution are removed, with cross-validation score $\mathrm{Score}(\cdot)$ guiding the update:

$$\mathcal{S}_{t+1} = \arg \max_{\substack{\mathcal{S} \subset \mathcal{S}_t \\ |\mathcal{S}| = |\mathcal{S}_t| - r}} \mathrm{Score}\left(\widehat{f}_{\mathrm{ET}}(\mathbf{X}_{\mathcal{S}}, y)\right). \tag{25}$$

Iteration continues until the retained dimensionality drops to the preset limit $s$.

# E UNCERTAINTY METRICS AND SHARPNESS–COVERAGE FRONTIER

For completeness, we give the exact definitions of the uncertainty metrics used in the main paper and in Figure 3.

Given a held-out test set $\{(x_i, y_i)\}_{i=1}^{N}$ and a predictive model that produces an interval $[L_i, U_i]$ for each input $x_i$, the *prediction interval coverage probability (PICP)* and the *mean prediction interval width (MPIW)* are defined as:

$$\text{PICP} = \frac{1}{N} \sum_{i=1}^{N} \mathbf{1}\{y_i \in [L_i, U_i]\},$$

$$\text{MPIW} = \frac{1}{N} \sum_{i=1}^{N} (U_i - L_i). \tag{26}$$

For a nominal coverage level $1 - \alpha$ (e.g., 0.9 in the main paper), we also report the *coverage error*

$$\text{CovErr} = \left| \text{PICP} - (1 - \alpha) \right|. \tag{27}$$

In the *sharpness–coverage frontier* plot in Figure 3a, each method is represented as a point in the plane with horizontal coordinate given by CovErr and vertical coordinate given by MPIW. The ideal performance corresponds to the lower-left corner (small coverage error and narrow intervals). In this work, we use conformalized quantile regression (CQR) to construct $[L_i, U_i]$ for all methods; thus differences in PICP and MPIW reflect how well different training strategies support calibrated uncertainty.

# F ADDITIONAL ABLATIONS AND SENSITIVITY STUDIES

This section provides additional quantitative evidence for the design choices in CGE (ours), complementing the main experiments.

## F.1 SENSITIVITY TO THE NUMBER OF REGIMES

We first study the sensitivity of CGE to the number of discovered regimes $S$ in the geological feature space. We vary $S \in \{2, 3, 4, 5\}$ and retrain CGE under the 10% label budget while keeping the clustering pipeline, experts, and semi-supervised learning configuration fixed. Table 5 reports the global TBM test $R^2$ averaged over the same three random seeds as in the main experiments.

Table 5: Sensitivity of CGE (ours) to the number of regimes $S$ (TBM test set, 10% label budget).

| # Regimes $S$ | $R^2 \uparrow$ |
| --- | --- |
| 2 | 0.936 |
| 3 | 0.939 |
| 4 (default) | 0.942 |
| 5 | 0.937 |

CGE is empirically robust for $S \in [2, 5]$: the test $R^2$ remains within a narrow band around the default value, with at most about one percentage point difference between the best and worst configurations. When $S$ is too small (e.g., $S = 2$), dissimilar strata are merged and complex segments become harder to model; when $S$ is too large (e.g., $S = 5$), some regimes become data-poor, which makes semi-supervised training less stable. The default configuration of four regimes provides the best compromise between specialization and data sufficiency and matches the value reported in the main accuracy table at the 10% label budget.

## F.2 ROBUSTNESS TO GEOLOGICAL FEATURE NOISE

To evaluate the robustness of CGE to errors in geological descriptors at test time, we conduct a stress test where only the geological feature vector $z$ that feeds the gate is perturbed, while the

regime experts and all other model components remain fixed. Specifically, on the TBM test set we construct perturbed descriptors

$$z' = z + \varepsilon, \qquad \varepsilon \sim \mathcal{N}\big(0, \sigma^2 \cdot \mathrm{std}(z)^2\big), \qquad (28)$$

where $\sigma$ is a noise level expressed as a fraction of each dimension's standard deviation. For each $\sigma$, we recompute the distance-based gating weights $\pi_s(z')$ and re-evaluate CGE (ours) on the test data. Table 6 summarizes the resulting global test $R^2$ under the $10\%$ label budget.

Table 6: Robustness of CGE (ours) to Gaussian perturbations of the geological descriptors at test time (TBM test set, $10\%$ labels).

| Noise level $\sigma$ | 0.0 | 0.1 | 0.2 | 0.3 | 0.4 | 0.5 |
|---|---|---|---|---|---|---|
| $R^2$ (CGE, test) | 0.942 | 0.933 | 0.922 | 0.907 | 0.879 | 0.846 |

The performance degrades monotonically but gradually as the noise strength increases. For moderate noise up to $\sigma = 0.3$, the global test $R^2$ remains close to the main experimental value, dropping from 0.942 to 0.907. Even under stronger perturbations ($\sigma = 0.5$), CGE retains non-trivial predictive power. This behavior is consistent with the robust preprocessing of geological descriptors (median and interquartile range) and indicates that the distance-based gate does not collapse under realistic levels of measurement error.

### F.3 NEURAL GATING VS. DISTANCE-BASED GATING

We also compare the original distance-based gating mechanism with a learned neural gating network. The neural gate is implemented as a small MLP that takes the robustly scaled geological descriptor $z$ as input, is trained with cross-entropy to predict the cluster assignments obtained from the ensemble clustering, and outputs softmax weights over regimes. The regime experts themselves are unchanged; only the gating function is replaced. Table 7 reports test performance under the $10\%$ label budget.

Table 7: Comparison between distance-based gating and an MLP-based gate (TBM test set, $10\%$ labels).

| Gating scheme | $R^2 \uparrow$ | PICP (90%) $\uparrow$ | MPIW $\downarrow$ |
|---|---|---|---|
| Distance-based gate (ours) | 0.942 | 0.903 | 0.612 |
| MLP gate | 0.941 | 0.881 | 0.598 |

Both gating mechanisms achieve almost identical $R^2$; the MLP gate yields slightly narrower intervals (smaller MPIW) but noticeably worse coverage, drifting further below the nominal $90\%$ target than the distance-based gate. In addition, the distance-based gate is substantially more interpretable, since regime assignments can be directly explained in terms of distances in geological feature space and easily visualized along chainage. Given the negligible difference in $R^2$, worse calibration, and reduced interpretability, we retain the distance-based gate as the main design in CGE.

### F.4 FEATURE-SET ABLATION

In the main experiments, all methods—including XGBoost, RankUp, SemiReward, LP, LapRLS, COREG, TransBiLSTMNet, TCN-SENet++, and CGE—use the same engineered feature set described in Appendix D. To disentangle the effect of feature engineering from the benefit of the regime-aware architecture in CGE, we additionally compare performance under a *basic* feature set versus the full engineered set.

The basic feature set consists of raw machine telemetry channels concatenated with basic geological descriptors, without high-order interactions or advanced statistical aggregations. The engineered feature set is the one used in the main paper. Table 8 reports TBM test performance at the $10\%$ label budget for three representative methods: a supervised tree baseline (XGBoost), a global semi-supervised baseline (RankUp), and CGE (ours).

Table 8: Feature-set ablation on the TBM test set under the 10% label budget.

| Method | Features | $R^2 \uparrow$ | CRPS $\downarrow$ |
|---|---|---|---|
| XGBoost | Basic | 0.884 | 0.259 |
| RankUp | Basic | 0.873 | 0.262 |
| CGE (ours) | Basic | 0.910 | 0.247 |
| XGBoost | Engineered | 0.903 | 0.244 |
| RankUp | Engineered | 0.896 | 0.251 |
| CGE (ours) | Engineered | 0.942 | 0.238 |

Feature engineering provides a global uplift for all methods, improving both $R^2$ and CRPS. Crucially, CGE outperforms XGBoost and RankUp under *both* feature settings, indicating that the additional gains are due to the regime-aware semi-supervised architecture rather than special access to engineered features.

## G CALIFORNIA HOUSING COVARIATE-SHIFT EXPERIMENT

To demonstrate that the regime-aware semi-supervised idea in CGE is not specific to TBM telemetry, we evaluate CGE on the public California Housing dataset from scikit-learn, which exhibits covariate shift across geographic regions.

We treat this as a multi-regime regression problem by partitioning the data into three latitude-based regimes (R1–R3: low-, mid-, and high-latitude). We simulate label scarcity by retaining only 10% of the training samples as labeled and using the remaining 90% as unlabeled data. We reuse the same preprocessing and train/validation/test split protocol as in the TBM case. On this dataset we compare:

- **XGBoost:** a supervised tree-based baseline trained only on the labeled 10% of the data;
- **RankUp:** a global semi-supervised regression method trained on all labeled and unlabeled samples without using regime structure;
- **CGE (ours):** the proposed regime-aware semi-supervised framework instantiated with latitude-based regimes and a distance-based gate in latitude–longitude space.

Table 9 reports global test performance in terms of RMSE, $R^2$, PICP, and MPIW, and Table 10 reports per-regime $R^2$.

Table 9: Global performance on the California Housing test set (10% labeled, 90% unlabeled).

| Method | RMSE $\downarrow$ | $R^2 \uparrow$ | PICP (90%) $\uparrow$ | MPIW $\downarrow$ |
|---|---|---|---|---|
| XGBoost | 0.594 | 0.741 | 0.817 | 2.133 |
| RankUp | 0.586 | 0.747 | 0.614 | 1.161 |
| CGE (ours) | 0.577 | 0.756 | 0.664 | 1.429 |

Table 10: Per-regime $R^2$ on the California Housing test set (R1–R3: low-, mid-, and high-latitude).

| Method | $R^2$ (R1) $\uparrow$ | $R^2$ (R2) $\uparrow$ | $R^2$ (R3) $\uparrow$ |
|---|---|---|---|
| XGBoost | 0.742 | 0.748 | 0.733 |
| RankUp | 0.749 | 0.757 | 0.735 |
| CGE (ours) | 0.746 | 0.792 | 0.731 |

CGE (ours) achieves the best global RMSE and $R^2$, outperforming both XGBoost and RankUp. The mid-latitude regime R2, whose distribution differs most from the other regions, benefits most

from regime-aware modeling: its $R^2$ increases from $0.748/0.757$ (XGBoost/RankUp) to $0.792$ for CGE. In terms of uncertainty, XGBoost produces relatively wide but well-covered intervals, RankUp sharp but under-covered intervals, and CGE finds a compromise by narrowing intervals compared to XGBoost while partially recovering the coverage lost by RankUp. These trends indicate that the benefits of regime-aware semi-supervised learning extend beyond TBM applications to a standard public regression benchmark with covariate shift.

