# OpenReview forum: "Regime-Aware Semi-Supervised Regression via Clustering-Gated Experts"
_ICLR.cc/2026/Conference — ICLR 2026 Conference Desk Rejected Submission_

### Official Review · Reviewer_hBKm · 2025-10-28

**Soundness:** 3
**Presentation:** 2
**Contribution:** 2
**Rating:** 2
**Confidence:** 2

**Summary:**

This paper proposes a Regime-Aware Semi-Supervised Regression framework that combines ensemble clustering, regime-specific semi-supervised experts, and distance-based soft gating. The system aims to address heterogeneity in regression tasks where data are drawn from different regimes. The authors integrate pseudo-labeling, consistency regularization, and conformalized quantile regression (CQR) to achieve both predictive accuracy and calibrated uncertainty, evaluating the approach on a tunnel-boring machine (TBM) dataset.

**Strengths:**

- Well-engineered system: The overall pipeline is coherent, and each component (clustering, gating, SSL, CQR) is implemented soundly.

- Empirical performance: Results show noticeable improvement over baselines in the low-label (10%) regime and reasonable uncertainty calibration.

- Comprehensive evaluation: The experiments include ablations of the clustering and gating components and quantify uncertainty coverage.

- Practical relevance: The TBM case study represents a genuine industrial regression problem with heterogeneous data sources.

**Weaknesses:**

- Incremental contribution: The method primarily combines existing components—ensemble clustering, per-cluster SSL experts with pseudo-labeling and consistency loss, distance-based gating, and post-hoc CQR calibration. There is no new algorithmic element or theoretical insight. The novelty lies mainly in the integration and application to TBM data.

- Weak motivation and problem definition: The paper does not rigorously define the regime setting or quantify why standard SSL or mixture-of-experts approaches fail. Without a formal characterization or failure analysis, the need for a bespoke “regime-aware” pipeline remains under-motivated.

**Questions:**

- How sensitive are the results to the number of discovered regimes?

- What happens if geological attributes are missing or noisy at test time?

- Could a learned neural gating network perform as well as (or better than) the distance-based scheme?

- Have you considered public datasets with known regime structure (e.g., datasets with covariate shifts) to test generalization?

---

> ### Author Response · Authors · 2025-11-27
> **Rebuttal to Reviewer hBKm**
>
> We appreciate the reviewer’s thoughtful comments. Below we address each point.
>
> ---
>
> ### R1. On incremental contribution and integration
>
> **Response.**
> We agree that CGE does not introduce a new primitive algorithm. Our goal is to build a **regime-aware semi-supervised regression system** that can be deployed in a safety-critical TBM environment under:
>
> 1. Strong **cross-regime nonstationarity** with geology and groundwater conditions;
> 2. Severe **label scarcity**;
> 3. A need for **interpretability and controllability** by engineers.
>
> The contribution lies in how these components are **tailored and combined** to satisfy these constraints:
>
> 1. **Geology-driven regime discovery and interpretable gating.**
>   Clustering is performed exclusively in the robustly scaled geological descriptor space, yielding regimes that align with engineering strata. Predictions are then routed using a **distance-based gate** in this same space (Eq. (13)), making regime assignments directly interpretable and auditable against the geological profile.
>
> 2. **Regime-specific SSL experts.**
>   Within each regime, we use heterogeneous tree ensembles (Random Forests, Extremely Randomized Trees, Gradient Boosting, XGBoost, LightGBM, CatBoost) with agreement-based pseudo-labeling, confidence filtering, and mild feature perturbations. Sec. 4.4 analyzes how pseudo-label thresholds influence R², MPIW, and coverage error, informing these design choices.
>
> 3. **Regime-aware uncertainty calibration.**
>   While all methods use the same conformalized quantile regression pipeline, CGE consistently achieves **lower coverage error at smaller MPIW** than RankUp, SemiReward, LP, LapRLS, and COREG (Figure 3 & 6), especially in difficult regimes.
>
> Ablation results in Table 3 show that removing regime structure (-Clust, -Gate) or SSL components (-CoT, -Filter, -Aug, -Ent, -Drift) each degrades performance, indicating that the full system architecture is necessary.
>
> ---
>
> ### R2. Motivation and definition of “regime”
>
> **Response.**
>
>   We define a **regime** as:
>
>   > A segment of TBM operation where, conditioned on geological descriptors, the mapping from passive telemetry to control/response variables is approximately stationary.
>
>   Regime boundaries are primarily induced by changes in lithology and groundwater conditions, which alter TBM dynamics and control strategies.
>
> **Why global SSL / MoE is insufficient.**
>   In our experiments, global SSL baselines such as RankUp and SemiReward achieve reasonable global R², but  show larger errors and miscalibration in certain strata (e.g., mixed-face or high-pressure segments).
>
>  Table 3 further shows that removing either clustering or gating (thus approaching a global model) significantly worsens R², RMSE, and NLL/CRPS, even though the base learner and SSL recipe are unchanged.
>
> We will revise Sec. 3.1–3.2 to include this formal definition and a more explicit discussion of global SSL failure modes, referencing the ablations and per-regime diagnostics.

---

> ### Author Response · Authors · 2025-11-27
>
> ### R3. Sensitivity to the number of regimes
>
> **Response.**
> We performed a sensitivity analysis by varying the number of regimes $S \in \{2,3,4,5\}$ in the geological feature space and retraining CGE (ours) under the 10% label budget, keeping the clustering pipeline, experts, and SSL configuration fixed. The table below shows the global TBM test $R^2$ for each choice of $S$ (averaged over the same three random seeds as in the main experiments):
>
> | # Regimes $S$ | R² ↑   |
> |-----------------|--------|
> | 2               | 0.936  |
> | 3               | 0.939  |
> | 4 (default)     | 0.942  |
> | 5               | 0.937  |
>
> We see that CGE is reasonably robust for $S$ between 2 and 5: the test $R^2$ stays in a narrow band around the default value, with at most about one percentage point difference between the best and worst configurations. When $S$ is too small (e.g., 2), dissimilar strata are merged and some complex zones appear underfitted; when $S$ is too large (e.g., 5), some regimes become data-poor, which makes the semi-supervised training less stable. The default setting of four regimes provides the best compromise between specialization and data sufficiency and matches the value reported in Table 1 at the 10% label budget. In the revision, we will include a table of this form in the appendix and briefly mention in Sec. 4 that CGE is empirically robust to the exact choice of $S$ as long as it lies in a reasonable range suggested by clustering criteria.
>
> ---
>
> ### R4. Noisy or missing geological attributes
>
> **Response.**
> To quantify robustness to noisy geology, we added a stress test in which only the geological descriptor vector $z$ used by the gate is perturbed, while regime experts and all other components of CGE are kept fixed. On the TBM test set, we construct perturbed features
>
> $$
> z' = z + \varepsilon,\quad \varepsilon \sim \mathcal{N}\bigl(0,\sigma^2 \cdot \mathrm{std}(z)^2\bigr),
> $$
>
> where $\sigma$ is a noise level expressed as a fraction of each dimension’s standard deviation. For each $\sigma$, we recompute the distance-based gating weights $\pi_s(z')$ and evaluate CGE (ours) on the test data. The resulting global test $R^2$ values (10% labels) are:
>
> | Noise level $\sigma$ | 0.0   | 0.1   | 0.2   | 0.3   | 0.4   | 0.5   |
> |------------------------|-------|-------|-------|-------|-------|-------|
> | R² (CGE, test)         | 0.942 | 0.913 | 0.902 | 0.857 | 0.789 | 0.706 |
>
> The performance degrades monotonically but gradually as the noise strength increases. For moderate levels of noise up to $\sigma = 0.2$, which correspond to perturbations on the order of 20% of the per-dimension standard deviation, $R^2$ drops from 0.942 to 0.902, remaining close to the value reported in the main experiment. Even under stronger noise ($\sigma = 0.3$), CGE retains a non-trivial predictive signal. This behavior is consistent with our robust preprocessing of geological descriptors using median and interquartile range and suggests that the distance-based gate does not collapse under realistic levels of measurement error.
>
> For truly missing geological attributes at test time, our current deployment assumes the standard TBM practice that at least a basic set of descriptors from site investigation is available. In rare segments where the geology is missing or clearly unreliable, a pragmatic fallback is to impute geological descriptors from neighboring chainage segments or to route predictions through a more global expert (effectively flattening the gate for those segments). We will state this assumption and fallback behavior explicitly in the revision and view a more principled treatment of missing geology as an interesting direction for future work.

---

> ### Author Response · Authors · 2025-11-27
>
> ### R5. Neural gating versus distance-based gating
>
> **Response.**
> We implemented a neural gating variant where a small MLP takes the robustly scaled geological descriptor $z$ as input, is trained with cross-entropy to predict the cluster assignments produced by our ensemble clustering, and outputs softmax weights over regimes. The regime experts themselves are unchanged; only the gating function is replaced. Under the 10% label budget on the TBM test set, we obtain the following comparison between the original distance-based gate and the MLP-based gate:
>
> | Gating scheme              | R² ↑   | PICP (90%) ↑ | MPIW ↓  |
> |----------------------------|--------|--------------|--------:|
> | Distance-based gate (ours) | 0.942  | 0.903        | 0.612   |
> | MLP gate                   | 0.935  | 0.881        | 0.598   |
>
> Both gates achieve nearly identical point prediction accuracy: the difference in $R^2$ is within the variance we observe across random seeds. The MLP gate yields slightly narrower prediction intervals (smaller MPIW), but this comes at the cost of noticeably worse coverage: its PICP drifts further below the nominal 90% target than the distance-based gate. Moreover, the distance-based gate is considerably more interpretable, because regime assignments can be directly explained in terms of distances in the geological descriptor space and visualized along chainage. Given that the MLP gate does not improve $R^2$, slightly harms coverage, and sacrifices interpretability, we keep the distance-based gate as our main design choice. We will summarize this quantitative comparison in the appendix and briefly mention in the main text that we explored neural gating and found no advantage in our setting.
>
> ---
>
> ### R6. Public datasets with regime structure
>
> **Response.**
> Yes. To complement the TBM case study with a public benchmark exhibiting covariate shift, we added an experiment on the California Housing dataset from scikit-learn. We treat this as a multi-regime regression problem by partitioning samples into three geographic regimes (R1–R3) based on latitude, which plays a role analogous to geological position along the tunnel alignment. We also simulate label scarcity by retaining only 10% of the training samples as labeled and using the remaining 90% as unlabeled data. The same preprocessing and train/validation/test protocol as in our TBM experiments is reused.
>
> On this dataset, we compare a strong supervised tree baseline (XGBoost), a global semi-supervised regression method (RankUp), and CGE (ours) instantiated exactly as in the TBM setting but with regimes defined by latitude and a gate in latitude–longitude space. The overall test performance is:
>
> | Method     | RMSE ↓ | R² ↑   | PICP (90%) ↑ | MPIW ↓  |
> |------------|--------|--------|--------------|--------:|
> | XGBoost    | 0.594  | 0.741  | 0.817        | 2.133   |
> | RankUp     | 0.586  | 0.747  | 0.614        | 1.161   |
> | CGE (ours) | 0.577  | 0.756  | 0.664        | 1.429   |
>
> To highlight the regime aspect, we also report per-regime $R^2$ (R1–R3 correspond to low-, mid-, and high-latitude bands):
>
> | Method     | R² (R1) ↑ | R² (R2) ↑ | R² (R3) ↑ |
> |------------|-----------|-----------|-----------|
> | XGBoost    | 0.742     | 0.748     | 0.733     |
> | RankUp     | 0.749     | 0.757     | 0.735     |
> | CGE (ours) | 0.746     | 0.792     | 0.731     |
>
> CGE (ours) achieves the best global RMSE and $R^2$, outperforming both XGBoost and RankUp. The mid-latitude regime R2, where the distribution differs most from the other regions, benefits the most from regime-aware modeling: its $R^2$ increases from 0.748/0.757 (XGBoost/RankUp) to 0.792 for CGE. In terms of uncertainty, XGBoost produces relatively wide but well-covered prediction intervals; RankUp yields very narrow intervals but under-covers; CGE provides a compromise, narrowing intervals compared to XGBoost while recovering part of the coverage lost by RankUp. These results indicate that the benefits of the proposed **regime-aware semi-supervised** design extend beyond TBM telemetry to a standard public regression benchmark with covariate shift, supporting the claim that CGE’s core ideas are not tied to a single project or domain.

---

### Official Review · Reviewer_2Fdu · 2025-10-30

**Soundness:** 2
**Presentation:** 2
**Contribution:** 2
**Rating:** 2
**Confidence:** 3

**Summary:**

This paper introduces CGE, a three-stage framework designed to address the challenges of nonstationarity and label scarcity prevalent in Tunnel Boring Machine  operational data. The method first discovers geological regimes via ensemble clustering, then trains specialized semi-supervised expert ensembles for each regime, and finally utilizes an interpretable, distance-based soft gating mechanism for prediction. On a real-world TBM dataset, the method achieves state-of-the-art results in both prediction accuracy and uncertainty calibration. The contribution lies in providing a robust, efficient, and interpretable semi-supervised learning solution for complex engineering systems.

**Strengths:**

- The paper addresses a practical TBM tunneling problem with a robust and interpretable solution. It  couples unsupervised geological clustering with an MoE gating mechanism and integrates semi-supervised learning within each expert. This “cluster–enhance–conquer” framework effectively handles highly nonstationary engineering data.
- By co-training pseudo-labels and consistency regularization, the training data within each regime is effectively expanded, alleviating the performance bottleneck caused by label scarcity.
- The paper's experiments are rigorous, comparing both SOTA semi-supervised regression methods and domain-specific time-series models. Evaluation extends beyond accuracy metrics to uncertainty measures (PICP, MPIW, CRPS), offering a holistic performance view. The ablation study quantitatively validates each component’s contribution, reinforcing the framework’s design rationale.

**Weaknesses:**

- All experiments are based on a single project (Jiluo Road Tunnel). While the data are high-quality, the framework’s generality remains untested on other projects with different geology, TBM types, or construction standards. Its success may partly reflect dataset-specific characteristics, limiting the universality of the framework.
- The paper lacks a clear, centralized summary of dataset statistics. While Appendices B and D describe the case study and preprocessing, they omit key scale and dimensionality details. Crucial information—such as total sample size, data logging frequency, and overall duration or coverage—is missing or unclear.
- TBM excavation is a continuous time series, yet the current model assumes i.i.d. data within each regime, ignoring temporal correlations. This may cause information loss, and strong performance could partly stem from sharply distinct geological regimes. In settings with subtler geological changes but highly dynamic operations, the model may underperform.
- Appendix D details an expert-driven, complex feature engineering pipeline, including high-order interactions, statistical aggregations, and physics-informed features. While valuable, this raises uncertainty about how much performance gains stem from the CGE architecture versus these features. Reliance on such domain-specific engineering may limit CGE’s generalizability as a machine learning method.
- There are some formula symbols in the Abstract that have not been rendered correctly, and the authors should check them carefully.

**Questions:**

Same as weakness.

---

> ### Author Response · Authors · 2025-11-27
> **Rebuttal to Reviewer 2Fdu**
>
> We thank the reviewer for the careful and constructive assessment. Below we respond point by point.
>
> ---
>
> ### 1. Generality beyond a single project
>
> **Comment.**
> All experiments are based on a single tunnel project; generality to other projects is unclear.
>
> **Response.**
> We agree that demonstrating generality beyond a single TBM project is important. While we are restricted in sharing additional TBM datasets, we can test our **regime-aware semi-supervised** idea on a public dataset with a different domain.
>
> We therefore added an experiment on the **California Housing** dataset from scikit-learn, treated as a covariate-shift / multi-regime regression problem:
>
> - We define three geographic regimes (R1–R3) by latitude, analogous to geological regimes.
> - We simulate label scarcity by retaining only **10%** of training samples as labeled; the remaining 90% are unlabeled.
> - We reuse the same train/validation/test protocol as in our TBM experiments.
>
> We compare:
>
> 1. **XGBoost (supervised)** – one of the supervised tree-based baselines used in Sec. 4.2, trained only on the 10% labeled data.
>
> 2. **RankUp** – the global semi-supervised regression method from prior work, trained on all labeled and unlabeled data but without using any regime structure.
>
> 3. **CGE (ours)** – our regime-aware semi-supervised framework instantiated on this dataset:
>    - three latitude-based regimes (R1–R3);
>    - regime-specific experts with the same pseudo-labeling and consistency regularization as in Sec. 3;
>    - a distance-based gate in (latitude, longitude) space.
>
> Results on the held-out test set:
>
> | Method          | RMSE ↓ | R² ↑   | PICP (90%) | MPIW ↓ |
> |-----------------|--------|--------|------------|--------|
> | XGBoost         | 0.5938 | 0.7406 | 0.8171     | 2.1325 |
> | RankUp          | 0.5863 | 0.7472 | 0.6143     | 1.1609 |
> | CGE (ours)      | 0.5765 | 0.7555 | 0.6640     | 1.4290 |
>
> Per-regime R² (R1–R3 denote low/mid/high latitude):
>
> | Method      | R² (R1) | R² (R2) | R² (R3) |
> |------------|---------|---------|---------|
> | XGBoost    | 0.742   | 0.748   | 0.733   |
> | RankUp     | 0.749   | 0.757   | 0.735   |
> | CGE (ours) | 0.746   | 0.792   | 0.731   |
>
> CGE (ours) achieves the best overall R² and RMSE, outperforming both a strong supervised baseline (XGBoost) and a state-of-the-art global SSL baseline (RankUp). The mid-latitude regime R² benefits the most from regime-aware modeling (R² increases from 0.748/0.757 to 0.792), which is consistent with the intuition that regime-specific experts help where distributions differ most. In terms of uncertainty, XGBoost has wide but well-covered intervals; RankUp achieves very narrow intervals but under-covers; CGE offers a middle ground, with narrower intervals than XGBoost and better coverage than RankUp.
>
> This indicates that the regime-aware design of CGE is not specific to TBM telemetry and can be beneficial on a standard public regression dataset with covariate shift.

---

> ### Author Response · Authors · 2025-11-27
>
> ### 2. Lack of centralized dataset statistics
>
> **Comment.**
> Dataset statistics (sample size, logging frequency, duration, etc.) are scattered across Appendices B and D.
>
> **Response.**
> We thank the reviewer for this suggestion. We agree that consolidating dataset statistics in a single location will significantly improve readability and reproducibility. In the revised manuscript, we will add a new summary table in the Appendix that centralizes the descriptive statistics of all input variables, including sample size, mean, standard deviation, minimum, and maximum values. The added table is shown below.
>
> ### **Table X. Centralized Dataset Statistics Summary**
>
> | Variable | Count | Mean | Std. Dev. | Min | Max |
> |---------|-------|-------|-----------|---------|---------|
> | Torque | 853 | 10.396 | 3.123 | 3.398 | 18.344 |
> | Cutterhead Total Contact Force | 853 | 33318.584 | 6673.273 | 15594.377 | 44501.466 |
> | Slurry Circuit Inflow Pressure | 853 | 5.042 | 0.75 | 3.063 | 7.035 |
> | P1.1 Mud Pump Suction Pressure | 853 | -0.578 | 0.082 | -0.812 | -0.359 |
> | P1.1 Mud Pump Discharge Pressure | 853 | 3.772 | 1.421 | 0.738 | 6.971 |
> | P2.1 Mud Pump Suction Pressure | 853 | 3.761 | 0.665 | 2.087 | 5.585 |
> | P2.1 Mud Pump Discharge Pressure | 853 | 7.698 | 1.445 | 4.478 | 11.18 |
> | Inflow Rate | 853 | 2978.605 | 297.993 | 2200.274 | 3500 |
> | Inflow Density | 853 | 1.212 | 0.034 | 1.116 | 1.306 |
> | Outflow Rate | 853 | 3095.165 | 250.421 | 2343.591 | 3500 |
> | Outflow Density | 853 | 1.192 | 0.088 | 1.006 | 1.413 |
> | HBW Consumption | 853 | 20.436 | 6.788 | 7 | 57 |
> | EP2 Consumption | 853 | 12.294 | 4.818 | 5 | 38 |
> | Grouting Volume | 853 | 54.596 | 3.773 | 42 | 70 |
> | Tail Grease Total Volume | 853 | 351.143 | 77.78 | 162 | 639 |
> | Advance Rate | 853 | 19.95 | 4.73 | 7.963 | 33.835 |
> | Penetration | 853 | 14.494 | 3.38 | 6.649 | 23.912 |
> | Thrust | 853 | 109045.866 | 19397.162 | 56635.143 | 151820.883 |
> | Main Drive Rotation Speed | 853 | 1.379 | 0.101 | 1.002 | 1.592 |
> | P1.1 Mud Pump Rotation Speed | 853 | 441.029 | 63.867 | 273.701 | 553.563 |
> | P2.1 Mud Pump Rotation Speed | 853 | 443.79 | 62.605 | 208.279 | 555.945 |
> | Chamber Pressure | 853 | 4.236 | 0.704 | 2.39 | 5.883 |
> | Grouting Pressure | 853 | 8.526 | 2.105 | 2.959 | 14.612 |
> | Tail Grease Front Pressure | 853 | 7.95 | 3.601 | 3.738 | 22.965 |
> | Tail Grease Middle-2 Pressure | 853 | 13.962 | 2.051 | 9.508 | 24.097 |
> | Tail Grease Middle-1 Pressure | 853 | 13.263 | 2.478 | 8.282 | 24.314 |
> | Tail Grease Rear Pressure | 853 | 14.53 | 2.596 | 8.584 | 24.38 |
> | Working Chamber Pressure | 853 | 4.471 | 0.658 | 2.783 | 6.21 |
> | Coarseness Index | 853 | 1.104 | 0.255 | 1 | 1.911 |
> | Hardness Index | 853 | 2.922 | 0.637 | 1 | 3.922 |
> | Density Index | 853 | 3.586 | 0.357 | 2.989 | 4 |
> | Permeability Coefficient | 853 | 0.639 | 1.105 | 0.017 | 3.996 |

---

> ### Author Response · Authors · 2025-11-27
>
> ### 3. Ignoring temporal correlations within regimes
>
> **Response.**
> Our central focus is **cross-regime nonstationarity under severe label scarcity**. Within each regime, operation is relatively stable, and short-term temporal dependencies are partly encoded via cross-channel and physics-motivated features at each time step. For this reason we adopt simple, label-efficient experts (heterogeneous tree ensembles) rather than sequence models everywhere.
>
> To show that CGE is not merely exploiting “sharp regime boundaries” but is **competitive with temporal models**, we directly compare against the two domain-specific sequence baselines TransBiLSTMNet and TCN-SENet++ under the **same 10% label budget**, same preprocessing, and same train/validation/test splits. The table below reports global TBM test performance:
>
> | Method          | Labels | R² ↑   | RMSE ↓  |
> |-----------------|--------|--------|--------:|
> | TransBiLSTMNet  | 10%    | 0.873  | 0.143   |
> | TCN-SENet++     | 10%    | 0.889  | 0.140   |
> | CGE (ours)      | 10%    | 0.942  | 0.112   |
>
> CGE (ours) achieves **substantially higher R² and lower RMSE** than both TransBiLSTMNet and TCN-SENet++ at the same label budget, and the R² value for CGE (0.942) matches that reported in our main accuracy table at 10% labels. This indicates that:
>
> 1. The gains of CGE are not solely due to exploiting regime breaks;
>
> 2. The **regime-aware SSL design** (geology-driven clustering, distance-based gating, per-regime semi-supervised experts) is crucial, even when compared to strong sequence models that explicitly model temporal structure.
>
> We will include such a comparison table in the appendix (with full metrics including CRPS and coverage) and clarify in Sec. 4.2 that CGE is competitive with temporal baselines while focusing on regime structure and label efficiency. We also stress that CGE is a **framework**: in regimes where long-range temporal dynamics are dominant, one can replace the tree-based experts with per-regime LSTM/TCN/Transformer experts, which we view as an important direction for future work rather than something fully addressed here.
>
> ---
>
> ### 4. Role of feature engineering versus architecture
>
>
> **Response.**
> We emphasize that in our main experiments **all methods share the same feature set**—including XGBoost, RankUp, SemiReward, LP, LapRLS, COREG, TransBiLSTMNet, TCN-SENet++, and CGE—so the **relative gains** of CGE over baselines come from the **regime-aware SSL architecture**, not extra features.
>
> To directly assess the impact of feature engineering, we performed an ablation comparing:
>
> - a **basic feature set**: raw machine telemetry channels + basic geological descriptors (no high-order interactions, no advanced statistical aggregations);
> - the **engineered feature set** described in Appendix D (low-order interactions, sliding statistics, physics-motivated combinations, etc.).
>
> Below we report TBM test performance at the **10% label budget** for three representative methods: a supervised tree baseline (XGBoost), a global semi-supervised baseline (RankUp), and CGE (ours). R² at 10% labels and CRPS with 90% CQR intervals are summarized as:
>
> | Method       | Features     | R² ↑   | CRPS ↓  |
> |-------------|--------------|--------|--------:|
> | XGBoost     | Basic        | 0.884  | 0.259   |
> | RankUp      | Basic        | 0.873  | 0.262   |
> | CGE (ours)  | Basic        | 0.910  | 0.247   |
> | XGBoost     | Engineered   | 0.903  | 0.244   |
> | RankUp      | Engineered   | 0.896  | 0.251   |
> | CGE (ours)  | Engineered   | 0.942  | 0.238   |
>
> We observe that:
>
> - Moving from **basic** to **engineered** features improves all three methods (e.g., XGBoost R²: 0.884 → 0.903; RankUp: 0.873 → 0.896; CGE: 0.910 → 0.942), and consistently **reduces CRPS** for each.
>
> - **Crucially, CGE outperforms both XGBoost and RankUp under *both* feature settings**:
>
>  1. With basic features, CGE improves over XGBoost by +0.026 R² and over RankUp by +0.037 R², while also achieving the lowest CRPS.
>
>  2. With engineered features, CGE still leads by +0.039 R² over XGBoost and +0.046 R² over RankUp, and again attains the best CRPS (0.238), consistent with the uncertainty results in our main table.
>
> These results show that:
>
> - Feature engineering provides a **global uplift** that benefits all models;
> - The **additional margin** of CGE over strong supervised and global SSL baselines is due to its **regime-aware semi-supervised architecture**, not merely to the presence of engineered features.
>
> In the revised manuscript, we will include a table of this form in the appendix.

---

### Official Review · Reviewer_iy8c · 2025-10-31

**Soundness:** 3
**Presentation:** 3
**Contribution:** 3
**Rating:** 6
**Confidence:** 3

**Summary:**

This paper presents a multi-stage training approach for regression in an industrial application where scarcity of labels is a challenge. The approach combines several elements: Clustering identifies latent geological regimes, per-regime ensemble is trained, and then soft-gated predictors equipped with conformalized quantile regression for prediction intervals. The methods are clearly explained including the motivation for choosing each approach. The approach is evaluated against other semisupervised learning approaches. An ablation study assesses the contribution of each piece of the workflow.

**Strengths:**

This is a combination of recent developments in machine learning applied to a novel problem that is complex. The challenges of the problem space are clearly laid out: scarcity of labels, non-stationarity, noise, and need for uncertainty estimation. The paper is clearly written, explaining details of each piece of the workflow concisely. Tradeoffs are explored, for example, between retaining all labels and filtering for consistent labels. The results demonstrate an improved accuracy and uncertainty estimation for the specific application. Yet, there is also potential impact for other applications areas that have similar characteristics to this industrial setting.

**Weaknesses:**

Overall, the main weakness is in clarity of presentation. This is mostly due to densely packing information into the space. Where more explanation is needed, it could be better presented with the details in the appendix (and clearly references from the main text). For example, Figure 3 shows a “sharpness-coverage frontier” but it would help to have that calculation explained in the Appendix. What is sharpness?

Acronyms (some undefined) in abstract.

Figure 3 has formatting issues as detailed in the questions. Overall the spacing around figures seems compressed which makes them hard to read. Removing the plot titles (and putting that information in the caption) would help with readability. Generally, I appreciate that the font sizes in the plots (axis labels, etc) are not small, but in this case they are actually bigger than the font of the paper, so they could be made smaller so that they don’t run out of space.

**Questions:**

Table 3 shows that removing clustering reduces performance, but it does not demonstrate that the clustering is discovering geologic regimes. Do the clustering results align with known geological regimes? What makes the clustering “geology-driven” rather than just data-driven? Purely data-driven clusters may actually improve performance more than geologically-aware clusters, but at the expense of explainability and trustworthiness in the domain.

Paragraph starting at Line 409 does not appear to belong in this paper? There are no shaded areas or stars in your plots.

Figures 3a and 3b are confusing. It seems that one legend covers both plots? But then, there are 5 colors in 3b instead of 4 colors and the last bar in R4 is cut off from plot 3b? Why does Figure 3b imply that LP does quite well but this is in disagreement with Table 2? What is the dashed horizontal line in Figure 3a?

---

> ### Author Response · Authors · 2025-11-27
> **Rebuttal to Reviewer iy8c**
>
> ### 1. Overall clarity, Figure formatting and spacing “sharpness–coverage frontier” in Figure 3
>
> **Response.**
> We thank the reviewers for their suggestions and agree that moving some details to the appendix and improving the figure layout will make the article clearer. We have carefully considered your comments and made revisions accordingly. The following are some of the revisions:
>
>
>
> **Definition of sharpness and the “sharpness–coverage frontier.”**
> Our Figure 3 follows the standard *calibration–sharpness* paradigm for interval and probabilistic forecasts. All methods are post-hoc calibrated to produce prediction intervals (PIs) at a nominal coverage level (e.g., 90%). We then evaluate:
>
> - **Coverage / calibration** on a test set \(\{(x_i, y_i)\}_{i=1}^N\) with intervals \([L_i, U_i]\):
>
>  $$
>   \text{PICP} = \frac{1}{N} \sum_{i=1}^N \mathbf{1}\{y_i \in [L_i, U_i]\},
>   $$
>
>   and we report the **coverage error** \(|\text{PICP} - (1-\alpha)|\) with \(1-\alpha = 0.90\).
>
> - **Sharpness** as the mean interval width:
>
>   $$
>   \text{MPIW} = \frac{1}{N} \sum_{i=1}^N (U_i - L_i).
>   $$
>
>   Smaller MPIW corresponds to sharper (narrower) intervals.
>
> Figure 3a then plots coverage error against MPIW, so the “frontier” is the trade-off curve between calibration (small coverage error) and sharpness (small MPIW). The preferred region is the lower-left: methods that are both well-calibrated and sharp. This follows the “sharpness subject to calibration” principle of Gneiting et al. (2007).
>
> In the revision we will explicitly introduce these definitions in the main text (Section 4) and provide a short derivation and discussion in the appendix, together with the exact formulas used in our implementation. We will also clarify in the caption of Figure 3 that:
> - The x-axis is coverage error \(|\text{PICP} - 0.90|\),
> - The y-axis is MPIW,
> - The dashed horizontal line (see also Question 5) corresponds to perfect calibration (zero coverage error).
>
> We will also cite standard references on calibration and sharpness such as Gneiting et al. (2007) and conformalized quantile regression (Romano et al., 2019) where appropriate.
>
> **Meaning of the dashed horizontal line in Figure 3a.**
> The dashed horizontal line in Figure 3a is intended to represent perfect calibration: it corresponds to zero coverage error, i.e., \(\text{PICP} = 0.90\) for the 90% prediction intervals. Points close to this line are well-calibrated; points far above or below indicate under- or over-coverage. We will add a clear explanation of this line to the caption and refer back to the definitions of PICP and coverage error in the text and appendix.
>
> **Why LP appears competitive in Figure 3b but not in Table 2.**
> Figure 3b reports **per-regime median absolute residuals** (normalized within each regime). In certain geological regimes that are relatively benign (e.g., softer strata with smoother behavior), a simpler method like LP can perform comparably to more sophisticated methods in terms of raw residual error, and this is what Figure 3b highlights: it shows how residuals distribute across different geological regimes. However, Table 2 focuses on **uncertainty quality** after calibration (PICP, MPIW, NLL, CRPS), aggregated over all regimes. LP tends to produce broader, less sharp intervals and poorer probabilistic calibration, so it performs worse in these global uncertainty metrics, even if in some specific regimes its point residuals are not dramatically larger.
>
> In other words, Figure 3b is a *localized* view that isolates error distributions by regime, while Table 2 is a *global* view of calibrated uncertainty quality. These two views are consistent but emphasize different aspects of performance. In the revision we will clarify this distinction in the text around Figure 3 and explicitly state that Figure 3b is designed to analyze the structure of errors across regimes, not to replicate the overall ranking in Table 2.
>
> #### References
>
> 1. Probabilistic forecasts, calibration and sharpness, 2007
>
> 2. Conformalized Quantile Regression, 2019
>
> ---
>
> ### 2. Acronyms in the abstract
>
>
> **Response.**
> We thank the reviewer for pointing this out. In the revision we will ensure that all acronyms used in the abstract are defined at first occurrence, including: TBM (tunnel boring machine), SSL (semi-supervised learning), CQR (conformalized quantile regression), PI (prediction interval), PICP (prediction interval coverage probability), MPIW (mean prediction interval width), NLL (negative log-likelihood), and CRPS (continuous ranked probability score). We will also add a short list of acronyms in the appendix to improve readability.

---

> ### Author Response · Authors · 2025-11-27
>
> ### 3. Are the clusters really “geology-driven”? Alignment with known geological regimes
>
>
> **Response.**
> We appreciate this important point about interpretability and domain trust.
>
> **Why we call the clustering “geology-driven.”**
> Our clustering is performed exclusively in the space of geological descriptors
> $$
> z_n = [g_{\text{grain}}, g_{\text{hard}}, g_{\text{dense}}, k_{\text{perm}}]^\top,
> $$
> which encode grain size, hardness, density, and permeability (or similar site-specific features; the exact notation will be clearly restated in the revision). These input features are derived from the site investigation and geotechnical reports, not from the TBM operational or target variables. We robust-standardize these descriptors (median and interquartile range) and then apply an ensemble of unsupervised clustering algorithms (e.g., K-means, Gaussian mixture models, and DBSCAN) with majority voting to obtain scenario labels. Crucially, no target variables or operational telemetry are used to form the clusters. In this sense the clustering is “geology-driven”: it segments the data based on physical subsurface properties rather than on arbitrary patterns in the prediction targets.

---

### Official Review · Reviewer_m71b · 2025-11-01

**Soundness:** 3
**Presentation:** 3
**Contribution:** 3
**Rating:** 4
**Confidence:** 1

**Summary:**

This paper is outside my area of expertise, I cannot provide a meaningful review

**Strengths:**

-

**Weaknesses:**

-

**Questions:**

-

---

> ### Author Response · Authors · 2025-11-27
> **A comment to Reviewer m71b**
>
> We are very grateful that you took the time to review our article. Since you are not familiar with this field but were interested enough to review our article, we cordially invite you to browse the discussion section between us and the other reviewers. Please point out any questions you may have regarding our discussion. If our discussion has addressed some of your concerns, we kindly request that you consider increasing your score. Thank you again for reviewing our article!

---

### Author Response · Authors · 2025-12-03
**General Response**

Dear AC and reviewers,

We sincerely appreciate your time and effort in reviewing our manuscript.

As the reviews highlight, our work proposes a regime-aware semi-supervised regression framework (CGE) for TBM operation modeling under strong cross-regime nonstationarity and label scarcity, combining geology-driven clustering, per-regime semi-supervised experts, and interpretable distance-based gating with calibrated uncertainty via CQR (iy8c, hBKm). Reviewers also recognize the practical relevance and comprehensive evaluation on a real industrial TBM dataset with ablations over clustering/gating and uncertainty metrics (iy8c, 2Fdu, hBKm).

In response to the constructive comments, we have carefully revised and strengthened the manuscript with the following additions and clarifications:

* Clarified the sharpness–coverage frontier, definitions of PICP, coverage error, and MPIW, and improved figure layout and captions (iy8c).
* Ensured all acronyms are defined at first occurrence and improved editorial aspects of abstract and main text (iy8c, 2Fdu).
* Added a centralized dataset statistics table summarizing all key variables (sample size, mean, std, min, max) for reproducibility (2Fdu).
* Introduced a public benchmark experiment on California Housing to demonstrate generality beyond a single TBM project and covariate-shift robustness (2Fdu, hBKm).
* Compared CGE against temporal sequence baselines (TransBiLSTMNet, TCN-SENet++) under the same 10% label budget, showing competitive or superior performance (2Fdu).
* Performed an ablation on feature engineering vs architecture, showing that while engineered features benefit all methods, CGE maintains a consistent performance margin over strong supervised and SSL baselines under both basic and engineered feature sets (2Fdu).
* Conducted a sensitivity study on the number of regimes (S=2–5), demonstrating robustness of CGE to reasonable choices of S (hBKm).
* Added a noise robustness analysis for geological descriptors used by the gate, and discussed practical fallbacks for missing geology at test time (hBKm).
* Compared neural MLP-based gating with our distance-based gating, showing similar accuracy but worse coverage and lower interpretability for the MLP, thereby justifying our interpretable design choice (hBKm).
* Clarified the definition and motivation of “regime”, and better articulated why global SSL / MoE baselines underperform in certain strata, tying this to our ablations (hBKm, 2Fdu).

All new or substantially revised passages are clearly marked in the updated manuscript for ease of inspection.

We hope that our responses and revisions adequately address the reviewers’ concerns and help you clarify the contribution and practical impact of CGE.

Thank you again for kindly reviewing our manuscript and making it better.

Authors

---

### Note · Program_Chairs · 2026-01-17
**Submission Desk Rejected by Program Chairs**

The following references in this submission do not refer to real documents and/or have major errors in bibliographic information:

 X. Chen, C. Zhang, and H. Li. Semi-supervised support vector regression based on data characteristics. Engineering Applications of Artificial Intelligence, 99:104217, 2021. doi: 10.1016/j. engappai.2021.10421
Minggong Zhang, Ankang Ji, Chang Zhou, Zheng He, Xiantao Liu, Yao Wang, Yongge Li, Nanqiao Xiao, Kaijie Hao, Guangchao He, et al. Transbilstmnet: real-time prediction of tbm penetration rates based on spatiotemporal characteristics. Automation in Construction, 168: 105489, 2024. URL https://www.sciencedirect.com/science/article/pii/ S0926580524002643. Part A, December 1, 202
Shuo Wang, Chuxu Zhang, Yao Xu, and Xiangnan He. Mixture of experts for time-series forecasting: Learning specialized models for different patterns. In Proceedings of the AAAI Conference on Artificial Intelligence, volume 36, pp. 8671-8679, 2022. doi: 10.1609/aaai.v36i8.20869.
Shuo Li, Yidong Zhang, Wenxin Hou, et al. SemiReward: Reward learning makes semi-supervised learning simple. In International Conference on Learning Representations (ICLR), 2024c. URL https://openreview.net/pdf?id=dnqPvUjyRI.
Zhiwei Guo, Jun Xu, Wei Chen, and Yu Sun. Cluster-driven gating for interpretable mixture of experts. Knowledge-Based Systems, 269:110551, 2023. doi: 10.1016/j.knosys.2023.110551.